# SEEKER: Semi-Supervised Knowledge Transfer for Query-Efficient Model Extraction

## Abstract

Model extraction attacks against neural networks aim at extracting models without white-box access to model internals and training datasets. Unfortunately, most existing methods demand an excessive number of queries (up to millions) to reproduce a functional substitute model, greatly limiting their real-world applicability. In this work, we propose a query-efficient model extraction attack that effectively distills knowledge from publicly available data. To this end, we introduce a semantic alignment approach that trains the substitute model without interacting with the victim model. The proposed approach optimizes the substitute model to learn a generalizable image encoding pattern based on semantic consistency of neural networks. We further propose a query generator that enhances the information density of generated queries by aggregating public information, thereby greatly reducing the query cost required for constructing the substitute model. Extensive experiments demonstrate that our method achieves state-of-the-art performance which improves query-efficiency by as much as 50× with higher accuracy. Additionally, our attack demonstrates the capability of bypassing most types of existing defense mechanisms.

## 1 Introduction

The past decade has witnessed tremendous progress made by Deep Neural Networks (DNNs) in achieving human-level performance in various fields of applications, such as medicine, finance, and autonomous driving. DNN models carry high commercial values and sensitive information from the secret training data. Consequently, in many real-world applications, DNN models are provided as a black box, where only the inputs to and the outputs of the models can be observed. Unfortunately, recent works (Barbalau et al., 2020; Truong et al., 2021) unveiled that DNN models are still vulnerable to model extraction attacks even if the adversary can only access the models in a black-box manner. In such attacks, the adversary can obtain a substitute model that emulates the functionality of the original victim model solely through querying the black-box model with unlabeled inputs. Using the substitute model, it is shown that the adversary can infer sensitive attributes of other users (Zhang et al., 2023), craft tailored adversarial samples aimed at the victim model (Wang et al., 2022), or even reconstruct the secret training data employed by the victim (Kahla et al., 2022). However, existing attacks primarily concentrate on enhancing the accuracy or transfer attack success rate (ASR) of the extracted model, while paying limited attention to query-efficiency of the model extraction process. An excessively large number of queries are used in these methods to extract a useful substitute model from the victim, leading to higher attack costs and an increased likelihood of encountering restrictions from the victim-side defense mechanisms.

Existing model extraction attacks either synthesize queries from completely random inputs or with the assistance of publicly available data, both demanding an excessive number of queries. On the one hand, it is obvious that optimizing a generative network to produce queries from random distribution requires a large query budget to converge (Truong et al., 2021; Kariyappa et al., 2021). On the other hand, even with the assistance of public datasets, existing attacks are still deemed query-inefficient due to two main reasons. First, as shown in Figure 1(a), traditional public dataset based attacks optimize the substitute model only through online interaction with the victim model. Second, most attacks lack an effective query generation process that constructs information-rich queries from the public data (Orekondy et al., 2019; Pal et al., 2020; Barbalau et al., 2020). As a result, even the most query-efficient method (Sanyal et al., 2022) demands over 3M queries to extract a model that can reach 88% accuracy on CIFAR-10. More details can be found in the appendix.

Figure 1: A comparison between our proposed attack and conventional model extraction attacks.

In this paper, we propose SEEKER, a query-efficient model extraction framework based on SEmi-supErvised public Knowledge transfER, as shown in Figure 1(b). To tackle the aforementioned challenges, we devise an offline stage that pre-trains the substitute model without incurring any query costs, significantly improving the query-efficiency. Specifically, we design a semantic alignment scheme that optimizes generalizable encoding layers without requiring interaction with the victim model. The scheme is based on an intriguing observation that purely enforcing semantic self-consistency enables the substitute model to demonstrate similar activation patterns to the victim model. Moreover, we propose a multi-encoder query generator to efficiently enhance the consistency between the substitute and the victim models via parallel processing of multiple public data. As a result, SEEKER elevates the query-efficiency of model extraction to unprecedented levels while preserving high accuracy and ASR when compared to state-of-the-art (SOTA) methods. Experimental results demonstrate that our attack reduces the query budget by more than $50\times$ for obtaining the same level of ASR compared with the SOTA methods. SEEKER can also extract a substitute model with a remarkable accuracy of 93.97% on CIFAR-10, surpassing the performance of the most accurate model stealing approach. Besides, our results indicate that both active and passive defenses against model extraction attacks may fall short in guaranteeing the security and safety of cloud-based MLaaS schemes. Our main contributions are summarized as follows.

- **Query-free self-supervised training**: To the best of our knowledge, our proposed semantic alignment scheme is the first self-supervised training procedure for model extraction, which increases the similarity between the substitute and victim models with zero query cost.
- **Query-efficient query generator**: We propose a multi-input autoencoder for query generation in model extraction attacks, which elevates the information density of query inputs through integrating public knowledge in the latent space.
- **Reproducible SOTA results**: Our attack significantly reduces the query budget and achieves higher accuracy and ASR than existing model extraction attacks. The implementation of our framework will be publicly available.

## 2 RELATED WORKS

### 2.1 MODEL EXTRACTION

Model extraction attacks aim at reproducing a victim model without access to the model internals. Although query-efficiency is a major concern in practical model extraction, many existing works focus only on simply improving accuracy without considering the query budget limit. For example, Black-Box Ripper (Barbalau et al., 2020) requires a large number of queries in the generative evolutionary strategy to produce a small population of training samples. Meanwhile, some works (Zhou et al., 2020; Truong et al., 2021) try to generate queries from noise vectors without the help of public data, and therefore can take millions of queries to reproduce the victim model. Among those query-efficient model extraction attacks, some works (Orekondy et al., 2019; Yu et al., 2020) assume annotations in public datasets. For example, the adaptive policy of Knockoff Nets (Orekondy et al., 2019) relies on labels to organize the public data into a hierarchical architecture for its proposed active learning approach. CloudLeak (Yu et al., 2020) adopts a supervised extraction strategy that requires labels to fine-tune the substitute. Recently, a new model extraction setting is explored, where the adversary only has access to some unlabeled public datasets. For example, Mosafi et al. (2019) generate query inputs by linearly merging public data. ActiveThief (Pal et al., 2020) attempts to select the most informative public data with active learning strategies. DFMS (Sanyal et al., 2022) crafts queries with a generative adversarial network (GAN), and utilizes the public datasets to assist the training process of GAN. Unfortunately, such attacks still lack query-efficiency, either due to inadequate information richness per query or lengthy generator pre-training.

### 2.2 BLACK-BOX ADVERSARIAL ATTACKS

An important application of model extraction is mounting black-box adversarial attacks. Generally speaking, we can classify black-box adversarial attacks into three categories: substitute-based,

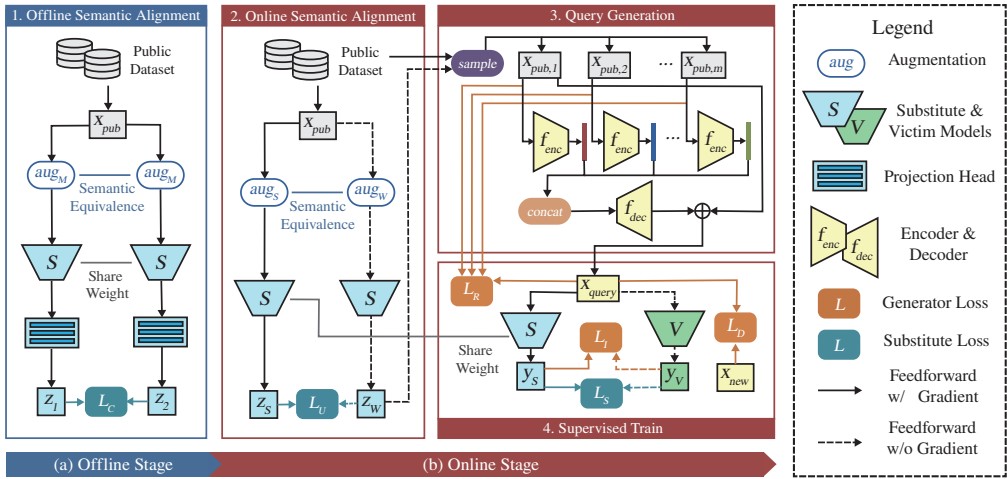

Figure 2: The overall framework of the proposed SEEKER. It consists of an offline pre-training stage and an online querying stage.

transfer-based, and query-based. First, a number of *substitute-based attacks* have already demonstrated the effectiveness of using the extracted substitute model as a base for launching black-box adversarial attacks (Papernot et al., 2017; Zhou et al., 2020; Wang et al., 2021). Additionally, *transfer-based attacks* assume the adversary can obtain a substitute model trained on the same dataset as the victim model, and focus on improving the transferability of the adversarial samples synthesized based on the substitute model (Inkawhich et al.; Wu et al., 2021; Zhang et al., 2022). Therefore, substitute-based and transfer-based attacks are generally complementary to each other. Finally, there are also *query-based attacks* (Li et al., 2019; Bai et al., 2020; Yuan et al., 2021) that directly utilize queries to construct adversarial samples. Most query-based adversarial attacks design optimization algorithms, such as gradient estimation methods (Tu et al., 2019; Ilyas et al., 2019), Bayesian optimization (Ru et al., 2019) or geometric mechanism (Maho et al., 2021), to construct adversarial samples. However, when the number of adversarial samples increases, such methods will also consume an impractically large number of queries. Some recent works (Ma et al., 2021; Yuan & He, 2021) incorporate substitute models into query-based approaches and achieve state-of-the-art query-efficiency among query-based attacks. However, such attacks are still only query-efficient when a very small number of adversarial samples are needed. Moreover, when query-based methods lose the connection to the API of the victim, they can no longer craft new adversarial samples.

## 3 METHOD

### 3.1 THREAT MODEL

Here, we formalize the threat model of model extraction attacks considered in this work. Given a victim model $V$ trained on a secret dataset $\mathcal{D}_{secret}$, an adversary attempts to extract a substitute model $S$ that mimics the behavior of $V$. The adversary can further generate perturbation $\boldsymbol{z}$ for a clean image $\boldsymbol{c} \in \mathcal{C}$ based on $S$ so that $\boldsymbol{c} + \boldsymbol{z}$ is misclassified by $V$. In particular, we note that the adversary is aimed at attaining high query-efficiency while retaining high accuracy and attack success rate (ASR). Here, we assume that the adversary can only obtain the output probability of $V$, i.e., the adversary has no access to the training dataset ($\mathcal{D}_{secret}$), the hyperparameters and weights of $V$. Similar to many previous works in model extraction (Orekondy et al., 2019; Pal et al., 2020; Barbalau et al., 2020), we make the assumption that the adversary has access to an unlabeled public dataset $\mathcal{D}_{pub}$, which is assumed to have a different distribution from $\mathcal{D}_{secret}$. Additionally, we assume that the adversary can prepare the attack in an offline stage and query $V$ in the online stage.

### 3.2 FRAMEWORK OVERVIEW

As shown in Figure 2, we propose a model extraction framework based on semi-supervised learning. To reduce the query cost of the model extraction process, we combine a *query-free self-supervised* learning scheme (procedures 1 and 2) and a *query-efficient supervised* approach (procedure 3 and 4). The self-supervised scheme, namely semantic alignment, optimizes the substitute model to be self-consistent on $\mathcal{D}_{pub}$, and does not require any query to $V$. For query-free self-supervised learning scheme, we develop offline semantic alignment that pre-trains $S$ to learn generalizable

features before interacting with the $V$, and online semantic alignment that assists the supervised approach during the iterative querying process. In the supervised approach, we focus on extracting more information with fewer queries. To this end, we develop a multi-encoder query generator that simultaneously processes several queries to synthesize an information-extracting query. Notably, we are the first to propose an offline stage for model extraction and develop self-supervised learning approach to pre-train the substitute model. Here, we present a brief outline of our framework, while the formal details are provided in the appendix. First, in the offline stage, we carry out semantic alignment procedure as follows.

1. **Offline Semantic Alignment**: In the offline semantic alignment process, the adversary pre-trains the substitute model $S$ on the public dataset $\mathcal{D}_{pub}$ using our proposed offline semantic consistency loss explained in the following section.

In the online stage, SEEKER iterates through the following three procedures to train the substitute model. Here, we take the $i$-th iteration with the query number of $n_i$ as an example.

2. **Online Semantic Alignment**: In the online semantic alignment process, the adversary first generates pseudo labels of the unannotated public data, and then calculates the online semantic consistency loss. The pseudo labels are also involved in sampling generator inputs from $\mathcal{D}_{pub}$.
3. **Query Generation**: Here, the adversary uses an aggregated query generator to construct a set of the query inputs $\{\boldsymbol{x}_{query}^{i,j} = G(\boldsymbol{x}_{pub,1}^{i,j}, ..., \boldsymbol{x}_{pub,m}^{i,j}) | \boldsymbol{x}_{pub,1}^{i,j}, ..., \boldsymbol{x}_{pub,m}^{i,j} \in \mathcal{D}_{pub}, j = 1, ..., n_i\}$.
4. **Supervised Train**: In the supervised training process, the adversary obtains the $i$-th query dataset $\mathcal{Q}_i = \{(\boldsymbol{x}_{query}^{i,j}, \boldsymbol{y}_{query}^{i,j} = V(\boldsymbol{x}_{query}^{i,j})) | j = 1, ..., n_i\}$ by querying the victim. Then the adversary updates the overall query dataset $\mathcal{Q} = \bigcup_{k=1}^{i} \mathcal{Q}_k$. The adversary calculates the supervised loss based on $Q$, and optimizes $S$ with the supervised loss and online semantic consistency loss. After substitute training, the query generator $G$ is optimized based on $S$ and $\mathcal{Q}_i$.

## 3.3 SEMANTIC ALIGNMENT

We propose a self-supervised scheme for the substitute model to acquire similar features to the victim model based on the public data. Our approach builds on the assumption that a well-trained victim model maintains semantic consistency, i.e. outputs similar representations for different images featuring the same object. Leveraging this semantic consistency as an additional prior, we propose a semantic alignment scheme that also aligns substitute model representations for semantically equivalent data. The semantically equivalent data are constructed by transforming the same public data with a combination of basic augmentations, such as horizontal flip and color jittering. These augmentations are aimed at simulating diverse environmental scenarios, such as varied camera angles or lighting conditions. With our approach, the substitute model demonstrates similar encoding patterns to the victim, as shown in Figure 3. We note that the semantic alignment learns generalizable features exclusively from the unannotated public data and does not require any additional query budget. Additionally, we have provided a detailed theoretical analysis in our appendix. We devise different variants of our semantic alignment scheme for both the offline and online model extraction procedures.

### 3.3.1 OFFLINE SEMANTIC CONSISTENCY

During the offline semantic alignment process, we employ a Siamese network architecture (He et al., 2020; Chen & He, 2021) to enhance the semantic consistency of the substitute model. Specifically, we employ two substitute models that share the same weights to process two sets of semantically equivalent data, and subsequently align the outputs of the two models. For any unlabeled public data $\boldsymbol{x}_{pub} \in \mathcal{D}_{pub}$, $S$ is trained with a NT-Xent loss (Chen et al., 2020):

$$\mathcal{L}_C = -\log \frac{\text{sim}(S(\text{aug}_{\text{M}}^1(\boldsymbol{x}_{pub})), S(\text{aug}_{\text{M}}^2(\boldsymbol{x}_{pub}))}{\sum_{\boldsymbol{x}'_{pub} \in \mathcal{D}_{pub}, \boldsymbol{x}'_{pub} \neq \boldsymbol{x}_{pub}} \text{sim}(S(\boldsymbol{x}_{pub}), S(\boldsymbol{x}'_{pub}))}, \tag{1}$$

where $\text{aug}_{\text{M}}^1(\cdot)$ and $\text{aug}_{\text{M}}^2(\cdot)$ are medium-level augmentations, and $\text{sim}(a, b)$ is a similarity function measuring the resemblance of $a$ and $b$. In the offline stage, we replace the fully connected layers of the substitute model $S$ with a projection head to obtain the latent representations of $\boldsymbol{x}_{pub}$ encoded by $S$. Intuitively, the loss function maximizes the similarity between the representations for differently augmented views of the same data point, while pushing away the the representations of different data points.

We utilize Grad-CAM to visually demonstrate the impact of our offline semantic alignment scheme, as illustrated in Figure 3. Here, Grad-CAM produces a heat map for an image, which highlights

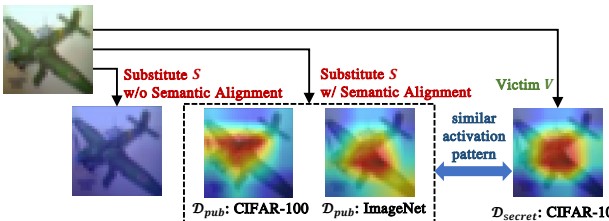
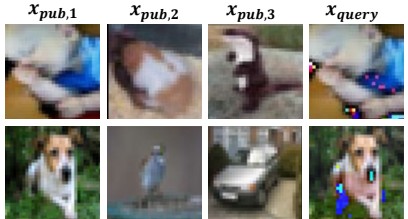

Figure 3: Comparisons of the activation heat maps between the substitute models with or without offline semantic alignment and the victim model.

Figure 4: Illustration of the queries synthesized by the aggregated query generator.

the crucial regions that a neural network is most activated to make predictions. We observe that the proposed training scheme enables $S$ to activate in a similar pattern to $V$ only using the public dataset $\mathcal{D}_{pub}$, even if $\mathcal{D}_{pub}$ follows a very different distribution (i.e., different classes of images) from the secret dataset $\mathcal{D}_{secret}$. Consequently, we see that adversaries can learn common encoding patterns from publicly available data, which can be utilized as *a-priori* knowledge for inferencing the properties of secret neural networks trained on private data.

### 3.3.2 ONLINE SEMANTIC CONSISTENCY

Different from the offline stage, the predictions of the substitute model becomes much closer to the victim model in the online stage. Based on this observation, we propose online semantic alignment that further improves the performance of the substitute model. Concretely, we first generate substitute output probabilities of weakly augmented public data. As the substitute model has similar predictions to the victim model, these probabilities can be used as pseudo labels for the public data. Then, we align the substitute outputs of strongly augmented data with the pseudo labels. Subsequently, we formulate the online semantic consistency loss as $\mathcal{L}_U = ||S(\mathrm{aug_W}(\boldsymbol{x}_{pub})), S(\mathrm{aug_S}(\boldsymbol{x}_{pub}))||_2$, where $\mathrm{aug_W}(\cdot)$ is weak augmentation and $\mathrm{aug_S}(\cdot)$ is strong augmentation.

### 3.4 AGGREGATED QUERY GENERATOR

To better leverage useful information from the public dataset under a limited number of queries, we propose an aggregated query generator that fuses multiple input data into a single information-extracting query. We consider three goals when designing the aggregated query generator: 1) **Aggregating**: the generator can effectively merge information from multiple public data, 2) **Informative**: the generator can produce information-extracting queries that minimize the gap between the substitute and the victim, 3) **Stealthy**: the synthesized queries maintain the structure of a natural image instead of collapsing into indistinguishable patterns. To achieve all three goals at the same time, we propose an aggregation architecture and three loss functions for the query generator.

### 3.4.1 AGGREGATION ARCHITECTURE

We design a multi-encoder network architecture to encode features from different public data. Concretely, to generate query input $\boldsymbol{x}_{query}$, the generator aggregates $m$ input data $\boldsymbol{x}_{pub,1}, ..., \boldsymbol{x}_{pub,m}$ from the public dataset. We can formulate the query generator $G$ as follows:

$$\boldsymbol{x}_{query} = G(\boldsymbol{x}_{pub,1}, ..., \boldsymbol{x}_{pub,m}) = f_{dec}([f_{enc}^1(\boldsymbol{x}_{pub,1}), ..., f_{enc}^m(\boldsymbol{x}_{pub,m})]) \oplus \boldsymbol{x}_{pub,1}, \quad (2)$$

where $f_{enc}^i(\cdot)$ denotes the $i$-th encoder, $f_{dec}(\cdot)$ the decoder, $[\cdot]$ the concatenation function, and $\oplus$ the element-wise addition operator. By applying Equation (2), we project the input data to the latent space with the respective encoders. Next, the decoder concatenates the representations of the public data and maps the latent code back to the image space. Finally, we apply a shortcut connection and add the first input $\boldsymbol{x}_{pub,1}$ to the output of the decoder. During the aggregation process, the generator regards $\boldsymbol{x}_{pub,1}$ as the base image and integrates the knowledge from the other public data $\boldsymbol{x}_{pub,2}, ..., \boldsymbol{x}_{pub,m}$ into $\boldsymbol{x}_{pub,1}$. Here, the multi-encoder design aligns with goal 1, and shortcut design goal 3. We have also included an alternative design of the aggregated architecture in the Appendix.

To sample more diversified public data as generator inputs, we design a sampling method based on the pseudo labels generated in online semantic alignment training step. In particular, we set the sampling probability for the $i$-th class as $p_i = \frac{-log_e(freq_i)}{\sum_{i=j}^{n_c} -log_e(freq_j)}$, where $freq_i$ denotes the frequency of the pseudo labels of the $i$-th class, and $n_c$ is the total number of classes.

### 3.4.2 Loss functions

We design the reconstruction loss, the inconsistency loss and the diversity loss to optimize the aggregated query generator.

1) **Reconstruction loss**: To fully aggregate different input data from the public dataset, we design the reconstruction loss to measure how well the query reconstructs the input data as:

$$\mathcal{L}_R = \frac{1}{m} \sum_{j=1}^{m} \alpha_j ||G(\boldsymbol{x}_{pub,1}, \boldsymbol{x}_{pub,2}, ..., \boldsymbol{x}_{pub,m}) - \boldsymbol{x}_{pub,j}||_2, \tag{3}$$

where $\alpha_j$ is a hyperparameter for balancing the stealthiness and information diversity in the generated query. We set $\alpha_1 = 1$ to preserve the basic appearance of $\boldsymbol{x}_{pub,1}$ in the generated query, and $0 < \alpha_2 = ... = \alpha_m \leq 1$ to ensure the generated query encodes information from $\boldsymbol{x}_{pub,j}$. Our reconstruction loss is aimed at achieving goal 1 and goal 3 at the same time.

We illustrate the synthesized queries of our aggregated query generator in Figure 4. In each group of the images, the generator $G$ merges $\boldsymbol{x}_{pub,1}$, $\boldsymbol{x}_{pub,2}$, and $\boldsymbol{x}_{pub,3}$ from $\mathcal{D}_{pub}$ to craft the query image $G(\boldsymbol{x}_{pub,1}, \boldsymbol{x}_{pub,2}, \boldsymbol{x}_{pub,3})$. As shown in Figure 4, the crafted query image largely resembles the original natural image $\boldsymbol{x}_{pub,1}$ from $\mathcal{D}_{pub}$, but has regional noises over some pixels that encode higher dimensional information from $\boldsymbol{x}_{pub,2}$ and $\boldsymbol{x}_{pub,3}$. Hence, we conclude that the aggregated query generator is effective in combining multiple input sources to produce information-rich queries that have similar visual patterns to natural images.

2) **Inconsistency loss**: The inconsistency loss can be formulated as:

$$\mathcal{L}_I = \exp(-\mathcal{L}_{KL}(S(G(\boldsymbol{x}_{pub,1}, \boldsymbol{x}_{pub,2}, ..., \boldsymbol{x}_{pub,m})), V(G(\boldsymbol{x}_{pub,1}, \boldsymbol{x}_{pub,2}, ..., \boldsymbol{x}_{pub,m})))), \tag{4}$$

where $\mathcal{L}_{KL}(\cdot)$ is the KL divergence. The main objective of the inconsistency loss is to optimize the aggregated generator such that the generator produces queries upon which the substitute and the victim models produce **different** prediction results. We point out that, in the online supervised learning stage, only those queries that cause $S$ to behave differently from $V$ are constructive in further training $S$ to be more similar to $V$.

3) **Diversity loss**: To craft more diversified query inputs, we introduce the diversity loss that reduces the similarity between new query inputs for the next iteration and existing query inputs. For the $i$-th iteration, the diversity loss can be formulated as

$$\mathcal{L}_D = \text{sim}(S(\boldsymbol{x}_{query}), S(G(\boldsymbol{x}_{pub,1}^{i+1}, \boldsymbol{x}_{pub,2}^{i+1}, ..., \boldsymbol{x}_{pub,m}^{i+1}))). \tag{5}$$

To reduce the computational complexity of optimizing over the diversity loss, we construct a dynamic diversity set $\mathcal{D}_{div}$ by selecting the most representative items from existing query inputs. Concretely, we add a query input to $\mathcal{D}_{div}$ if and only if the cosine distances between this query input and existing items in $\mathcal{D}_{div}$ are all above a threshold $T_{div}$. The inconsistency loss and diversity loss are proposed to fulfill goal 2.

The overall loss $\mathcal{L}_{Gen}$ for training the generator can be formulated as $\mathcal{L}_{Gen} = \mathcal{L}_R + \lambda_I \mathcal{L}_I + \lambda_D \mathcal{L}_D$, where $\lambda_I$ and $\lambda_D$ are hyperparameters to determine the relative importance of each loss item.

### 3.5 Supervised Training

Based on the query samples crafted in Section 3.4 and losses derived in Section 3.3.2, the overall loss for optimizing the substitute model can be formulated as $\mathcal{L}_{Sub} = \mathcal{L}_S + \lambda_U \mathcal{L}_U$. Here, $\mathcal{L}_U$ is the online semantic consistency loss, $\mathcal{L}_S$ is the supervised loss defined as $\mathcal{L}_S = \mathcal{L}_{KL}(S(\boldsymbol{x}_{query}), \boldsymbol{y}_{query})$, and $\lambda_U$ is a hyperparameter to balance the loss terms.

To avoid the substitute model being overfitted in every iteration, we combine two simple yet effective approaches: weighted query sampling and loss-based training termination. First, weighted query sampling is proposed to balance the importance of old and new query datasets, where queries in the $i$-th iteration are assigned with weight $w = \alpha^{-i}$, where $0 < \alpha < 1$. Second, we design a loss-based termination mechanism to automatically stop the substitute training process. Within each iteration, substitute training ceases when the average loss does not drop for several consecutive epochs.

## 4 Experiments

### 4.1 Experimental setting

**Datasets and models.** We use CIFAR-10, CIFAR-100 (Krizhevsky et al., 2009), Tiny ImageNet and ImageNet (Deng et al., 2009) datasets in our experiments, which are widely adopted by recent model

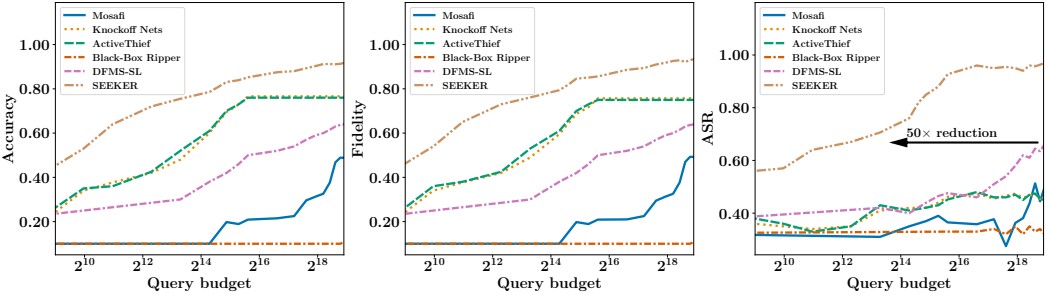

Figure 5: Accuracy, fidelity, and ASR comparisons between SEEKER and the SOTA model extraction attacks under different query budgets.

Table 1: Accuracy, fidelity, and ASR of different model extraction attacks under a relatively low query budget. The mean accuracy, fidelity, and ASR along with the standard deviations are provided.

| $\mathcal{D}_{secret}$ | $\mathcal{D}_{pub}$ | Attack | Acc (%) | Fid (%) | ASR (%) |
|---|---|---|---|---|---|
| CIFAR-10 (Victim Acc = 95.52%) | CIFAR-100 | Mosafi *et al.* | 26.19 ($\pm$1.32) | 26.15 ($\pm$1.29) | 32.98 ($\pm$2.78) |
| | | Knockoff Nets | 75.66 ($\pm$1.07) | 76.56 ($\pm$1.09) | 46.89 ($\pm$3.24) |
| | | ActiveThief | 75.23 ($\pm$0.92) | 76.41 ($\pm$0.93) | 45.04 ($\pm$3.38) |
| | | Black-Box Ripper | 10.72 ($\pm$1.00) | 11.06 ($\pm$0.98) | 34.91 ($\pm$2.58) |
| | | DFMS-SL | 52.57 ($\pm$1.13) | 53.34 ($\pm$1.13) | 45.93 ($\pm$2.43) |
| | | SEEKER (ours) | **88.01 ($\pm$0.97)** | **88.94 ($\pm$0.98)** | **96.43 ($\pm$2.69)** |
| | Tiny ImageNet | Mosafi *et al.* | 24.47 ($\pm$0.88) | 24.55 ($\pm$0.89) | 30.32 ($\pm$2.76) |
| | | Knockoff Nets | 83.66 ($\pm$1.12) | 84.28 ($\pm$1.08) | 50.63 ($\pm$3.01) |
| | | ActiveThief | 84.07 ($\pm$1.04) | 84.96 ($\pm$1.07) | 51.80 ($\pm$3.38) |
| | | Black-Box Ripper | 11.06 ($\pm$0.92) | 11.41 ($\pm$1.06) | 33.49 ($\pm$2.52) |
| | | DFMS-SL | 54.35 ($\pm$0.95) | 55.83 ($\pm$0.97) | 47.93 ($\pm$2.79) |
| | | SEEKER (ours) | **88.93 ($\pm$0.94)** | **89.26 ($\pm$0.85)** | **97.20 ($\pm$2.62)** |
| CIFAR-100 (Victim Acc = 78.72%) | CIFAR-10 | Mosafi *et al.* | 2.52 ($\pm$0.04) | 2.46 ($\pm$0.03) | 44.16 ($\pm$2.98) |
| | | Knockoff Nets | 54.88 ($\pm$0.91) | 55.98 ($\pm$0.98) | 48.67 ($\pm$2.86) |
| | | ActiveThief | 53.28 ($\pm$0.96) | 54.53 ($\pm$0.92) | 48.55 ($\pm$2.45) |
| | | Black-Box Ripper | 1.24 ($\pm$0.05) | 1.58 ($\pm$0.05) | 53.55 ($\pm$2.30) |
| | | DFMS-SL | 34.51 ($\pm$1.28) | 37.15 ($\pm$1.26) | 39.26 ($\pm$2.42) |
| | | SEEKER (ours) | **60.04 ($\pm$1.39)** | **63.81 ($\pm$1.42)** | **87.25 ($\pm$2.97)** |
| | ImageNet | Mosafi *et al.* | 3.88 ($\pm$0.05) | 3.91 ($\pm$0.07) | 44.03 ($\pm$2.53) |
| | | Knockoff Nets | 61.70 ($\pm$1.14) | 65.94 ($\pm$1.17) | 71.05 ($\pm$3.49) |
| | | ActiveThief | 62.68 ($\pm$1.08) | 66.25 ($\pm$1.09) | 72.50 ($\pm$3.70) |
| | | Black-Box Ripper | 1.33 ($\pm$0.03) | 1.69 ($\pm$0.04) | 49.95 ($\pm$2.83) |
| | | DFMS-SL | 37.32 ($\pm$1.10) | 40.89 ($\pm$1.17) | 45.41 ($\pm$3.08) |
| | | SEEKER (ours) | **72.23 ($\pm$1.01)** | **75.81 ($\pm$1.03)** | **95.35 ($\pm$2.72)** |

extraction methods (Barbalau et al., 2020; Truong et al., 2021; Sanyal et al., 2022). We use ResNet-34 (He et al., 2016) as the model architecture of the victim. To evaluate the performance of different attacks across diverse model architectures, we test four widely-used classical model architectures for $S$: ResNet-34 (He et al., 2016), PyramidNet (Han et al., 2017), DenseNet (Huang et al., 2017), and WRN-28 (Zagoruyko & Komodakis, 2016). In Figure 5, Table 2, Table 3 and Table 4, we use CIFAR-10 as $\mathcal{D}_{secret}$ and CIFAR-100 as $\mathcal{D}_{pub}$.

**Evaluation metrics.** Following existing methods, we use three metrics to evaluate model extraction attacks: accuracy (Acc), fidelity (Fid), and attack success rate (ASR). On top of the traditional prediction accuracy metric, we use fidelity (Jagielski et al., 2020) to measure how well the predictions of the substitute match with that of the victim (including both correct and incorrect predictions). Given a clean dataset $\mathcal{C}$, fidelity can be formulated as $\mathrm{Fid} = \frac{1}{|\mathcal{C}|} \sum_{\boldsymbol{c} \in \mathcal{C}} \mathbb{1}(V_l(\boldsymbol{c}) = S_l(\boldsymbol{c}))$, where $\mathbb{1}(\cdot)$ denotes the indicator function. Since an important application of model extraction is launching substitute-based adversarial attacks, we use ASR to measure the success of non-targeted black-box adversarial attacks, which is formulated as $\mathrm{ASR} = \frac{1}{|\mathcal{C}_V|} \sum_{\boldsymbol{c} \in \mathcal{C}_V} \mathbb{1}(V_l(\boldsymbol{c} + \boldsymbol{z}) \neq V_l(\boldsymbol{c}))$, where $\mathcal{C}_V$ is the dataset correctly classified by $V$. To compare query-efficiency of black-box adversarial attacks, we introduce query-efficiency ratio (QER) that measures the number of successful attacks per query as $\mathrm{QER} = \frac{1}{n_q} \sum_{\boldsymbol{c} \in \mathcal{C}_V} \mathbb{1}(V_l(\boldsymbol{c} + \boldsymbol{z}) \neq V_l(\boldsymbol{c}))$, where $n_q$ denotes the query number.

## 4.2 COMPARISONS WITH MODEL EXTRACTION ATTACKS

To demonstrate the effectiveness of our proposed framework, we compare SEEKER against existing model extraction attacks.

**Query-efficiency.** We first compare the query-efficiency between our attack and five SOTA model extraction attacks, including the attack proposed by Mosafi *et al.*(Mosafi et al., 2019), Knockoff Nets (Orekondy et al., 2019), ActiveThief (Pal et al., 2020), Black-Box Ripper (Barbalau et al., 2020), and DFMS-SL (Sanyal et al., 2022). For ASR comparisons, we use the white-box adversarial attack MI-FGSM (Dong et al., 2018) to perform non-targeted attacks over all model extraction methods. To ensure fair comparisons, the same set of parameters is employed for MI-FGSM across all the model extraction attacks in each experimental configuration.

As illustrated in Figure 5, SEEKER achieves a high level of accuracy, fidelity, and ASR with an extremely small query budget. We point out that our method can reduce the query budget by $5\times$ to achieve 75.7% accuracy, and by more than $50\times$ to achieve 65.5% ASR when compared to the SOTA methods. It is noteworthy that Knockoff Nets and ActiveThief only sample query inputs from the public dataset, thus reaching the optimal extraction performance when the query budget is equal to the size of the public dataset. In contrast, the performance of our attack continues to rise with further querying. We further demonstrate the accuracy, fidelity, and ASR of the aforementioned model extraction attacks under a relatively small query budget of 100K in Table 1. We note that, since Black-Box Ripper and DFMS-SL require millions of queries in the query generation process, their performance under small query budgets is relatively poor. On the other hand, although Knockoff Nets and ActiveThief can obtain a relatively high accuracy with a small number of queries, its attack success rate can be less satisfactory. In contrast, SEEKER achieves high accuracy, fidelity, and ASR across different datasets. For example, SEEKER attains 12.35% higher in accuracy and 49.54% higher in ASR than Knockoff Nets when using CIFAR-10 as $\mathcal{D}_{secret}$ and CIFAR-100 as $\mathcal{D}_{pub}$. We have included more results across different public datasets in the Appendix.

**Best achievable accuracy.** While our attack achieves remarkable query-efficiency compared to SOTA model extraction attacks, we note that the best achievable accuracy is an important metric for evaluating model extraction attacks, especially when the adversary has the capability of querying the victim model with an unlimited number of queries. Therefore, we perform different model extraction attacks under a significantly larger query budget to compare their highest attainable accuracy. As shown in Table 2, SEEKER achieves the highest accuracy amongst the SOTA model extraction attacks. Notably, our proposed attack attains 93.97% accuracy with a cost of 4M queries, whereas DFMS-SL requires 20M queries to extract a substitute of 93.96% accuracy.

**Model architecture generalization.** We further compare the accuracy of different model extraction attacks using diverse model architectures for the substitute model. As shown in Table 3, SEEKER exhibits better generalization capability across different substitute model architectures than the other attacks based on public datasets. The results demonstrate that our approach can effectively extract a substitute even if the victim and substitute models do not have the same architecture. We also provide a more detailed analysis regarding model architecture generalization in the appendix.

## 4.3 COMPARISONS WITH QUERY-BASED ATTACKS

We compare query-efficiency of our method against three query-based adversarial attacks: NES (Ilyas et al., 2018), Bandits (Ilyas et al., 2019), and Simulator attack (Ma et al., 2021) (SOTA). Figure 6 demonstrates QER of different black-box adversarial attacks by attacking 10K clean data from $\mathcal{C}$ for reaching similar levels of ASR with similar noise levels. Although the query-based attacks have higher QER for crafting a small number of adversarial samples, such attacks are easily outperformed by SEEKER when more than 1,600 samples are required. Furthermore, SEEKER is $1.9\times$ more query-efficient than Simulator attack when crafting adversarial 3000 samples, and $3.1\times$ as crafting 5,000 samples. Lastly, from Figure 6, we see that adversarial attacks based on substitute mod-

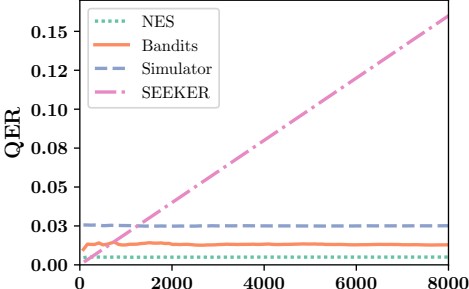

Figure 6: Query-efficiency comparisons between our method and query-based adversarial attacks.

els (e.g., SEEKER) achieve asymptotically higher query-efficiency than query-based attacks when the number of successful adversarial samples increases. We include more comparisons between

Table 2: Optimal accuracy of different attacks.

| Attack | Acc (%) |
|---|---|
| Knockoff Nets | 75.66 |
| Black-Box Ripper | 90.00 |
| DFMS-SL | 93.96 |
| SEEKER (ours) | **93.97** |

Table 3: Accuracy and ASR of different model extraction attacks with diverse model architectures.

| Architecture | ResNet-50 | | PyramidNet | | DenseNet | | WRN-28 | |
|---|---|---|---|---|---|---|---|---|
| Metric | Acc (%) | ASR (%) | Acc (%) | ASR (%) | Acc (%) | ASR (%) | Acc (%) | ASR (%) |
| Knockoff Nets | 75.66 | 46.89 | 77.24 | 71.43 | 66.13 | 46.84 | 77.82 | 29.52 |
| SEEKER (ours) | **88.01** | **96.43** | **87.43** | **91.10** | **87.32** | **84.74** | **88.56** | **96.73** |

black-box adversarial attacks based on our method and query-based adversarial attacks in the appendix.

## 4.4 ABLATION STUDIES

We carefully designed a set of ablation experiments to examine the contributions of each of the components in our framework. Table 4 confirms that both semantic consistency based unsupervised training and aggregated query generator contribute to the overall performance of SEEKER. In particular, the offline unsupervised training procedure and our aggregated query generator

Table 4: Ablation experiment for key contributions in our attack.

| Component | Acc (%) | Fid (%) | ASR (%) |
|---|---|---|---|
| Baseline (Random) | 74.30 | 75.87 | 46.92 |
| Baseline+Offline Semantic Alignment | 85.11 | 86.26 | 84.74 |
| Baseline+Online Semantic Alignment | 76.45 | 77.27 | 46.88 |
| Baseline+Aggregated query generator | 84.75 | 86.23 | 94.33 |
| SEEKER (ours) | **88.01** | **88.94** | **96.43** |

contribute to a 10.7% rise in accuracy and a 47.41% rise in ASR, respectively. Overall, the combination of the proposed techniques improved accuracy by as much as 13.71% and ASR by 49.51%.

## 4.5 PENETRABILITY AGAINST DEFENSE MECHANISMS

In this section, we evaluate the effectiveness of SEEKER against typical defense mechanisms, including active and passive approaches. We also provide a more detailed analysis in the appendix.

**Active defenses.** Typical active defenses against model extraction include adding perturbations (Sha et al., 2023), truncating the top-k outputs (Orekondy et al., 2019) and rounding output scores (Tramèr et al., 2016). Experimental results show that perturbation-based defense, while capable in reducing the performance of our attack, can also compromise the accuracy of the original victim model. For example, when Gaussian noise with a mean of 0 and a standard deviation of 0.5 is applied, the accuracy of our attack is decreased by 19%, accompanied with a significant 32% reduction in the accuracy of the victim model. For truncation and rounding, we consider an extreme setting where only the hard label is released by the victim, and show that our method can still achieve $0.92\times$ of the original accuracy under this setting. Although active defensive approaches can degrade the performance of our attack by a small degree, we note that altering the prediction scores also limits the utility of the victim model for honest users, as discussed in (Chandrasekaran et al., 2020).

**Passive defenses.** Existing passive defenses recognize model extraction attacks by analyzing the distribution of the query data. As a typical passive defense, PRADA (Juuti et al., 2019) computes the Shapiro-Wilk test statistic $W(D)$ to measure how the query input distribution deviates from the normal distribution. If $W(D)$ is below a threshold $\delta$, PRADA determines $D$ is from a model extraction attack. We set $\delta = 0.90$ in our experiments based on the original paper. Under a query budget of 100K, $W(D)$ for the query input distribution $D$ generated by SEEKER is 0.95, which is well above the threshold. The experimental results show that the distribution of the queries generated by our attack is only slightly deviated from normal distribution, and is able to circumvent PRADA detection. We point that the penetrability against distribution-based defense agree with the observation that the crafted queries mostly follow the distribution of a natural image, as demonstrated in Figure 4.

## 5 CONCLUSION

In this paper, we propose a query-efficient model extraction framework based on two-stage semi-supervised public knowledge transfer. Our key insight is that unannotated public datasets can be of great help to query-efficient model extraction. In particular, public data can be used in both unsupervised substitute training and informative query generation. By carefully designing the overall architecture of the framework, we show that SEEKER is able to significantly outperform the SOTA model extraction techniques in terms of accuracy, ASR, and query-efficiency.

## 6 REPRODUCIBILITY STATEMENT

The models and datasets for reproducing our results are introduced in the main manuscript, and more detailed experimental configurations can be found in the appendix. We point out that the datasets involved in our experimental evaluation are all publicly accessible. We also provide the code for SEEKER in our supplementary materials for better reproducibility. Please refer to the README file under the root directory for the introduction to our directory layout and detailed procedures to reproduce our results.

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

## A  COMPARISON WITH EXISTING MODEL EXTRACTION ATTACKS

In Table A1, we assess and compare the state-of-the-art model extraction attacks from various aspects. First, we observe that model extraction is a technique that is widely applied in model stealing and adversarial attacks, with some works that simultaneously try to perform both attacks using the extracted model. Second, in spite of the importance of query-efficiency in launching real-world attacks, we find that many existing methods require an impractically large amount of queries to obtain a useful substitute model. For instance, in order to reach 88% accuracy on CIFAR-10 (Krizhevsky et al., 2009), DFMS (Sanyal et al., 2022) needs to consume more than 3M queries, while our proposed method only requires 0.1M queries. Third, also as a way to improve query-efficiency, some works (Orekondy et al., 2019; Barbalau et al., 2020; Jagielski et al., 2020; Yu et al., 2020; Pal et al., 2020) seek to utilize public datasets as surrogate information in extracting the victim model. Nonetheless, many works explore only a small subset of the data points in public datasets due to the limited number of query budget (Pal et al., 2020; Yu et al., 2020), or assume that the public dataset is labeled (Orekondy et al., 2019; Yu et al., 2020). Lastly, the lack of open-source code for certain attack methods can limit their reproducibility.

Table A1: Qualitative comparisons between different model extraction attacks.

| Methods | Goal | Query-efficient | Public dataset | Open-source |
|---|---|---|---|---|
| Papernot *et al.* (Papernot et al., 2017) | Both | ✗ | No | ✓ |
| Mosafi *et al.* (Mosafi et al., 2019) | Model Stealing | ✗* | Unlabeled | ✓ |
| Knockoff Nets-Random (Orekondy et al., 2019) | Model Stealing | ✓ | Unlabeled | ✓ |
| Knockoff Nets-Adaptive (Orekondy et al., 2019) | Model Stealing | ✓ | Labeled | ✗ |
| CloudLeak (Yu et al., 2020) | Model Stealing | ✓ | Labeled | ✗ |
| Black-Box Ripper (Barbalau et al., 2020) | Model Stealing | ✗ | Unlabeled | ✓ |
| Jagielski *et al.* (Jagielski et al., 2020) | Both | ✗ | Unlabeled | ✗ |
| ActiveThief (Pal et al., 2020) | Model Stealing | ✓ | Unlabeled | ✓ |
| DaST (Zhou et al., 2020) | Adversarial attack | ✗ | No | ✓ |
| MAZE (Kariyappa et al., 2021) | Model Stealing | ✗ | No | ✗ |
| Wang *et al.* (Wang et al., 2021) | Adversarial attack | ✗ | No | ✗ |
| Truong *et al.* (Truong et al., 2021) | Model Stealing | ✗ | No | ✓ |
| DFMS (Data-free) (Sanyal et al., 2022) | Model Stealing | ✗* | No | ✓ |
| DFMS (Proxy data) (Sanyal et al., 2022) | Model Stealing | ✗* | Unlabeled | ✓ |
| SEEKER (Ours) | Both | ✓ | Unlabeled | ✓ |

\* We note that, while Mosafi *et al.* (Mosafi et al., 2019) and Sanyal *et al.* (Sanyal et al., 2022) proposed techniques to reduce the number of queries, the techniques are still not quite query-efficient compared to other methods.

## B  DETAILED ALGORITHM

A formalized algorithm of our proposed framework is shown in Algorithm 1. We describe the main steps as follows:

1. **Offline Semantic Alignment**: In the offline semantic alignment process, the adversary pre-trains the substitute model $S$ on the public dataset $\mathcal{D}_{pub}$ using our proposed offline semantic consistency loss (row 1).

In the online stage, SEEKER iterates through the following three procedures to train the substitute model. Here, we take the $i$-th iteration as an example.

2. **Online Semantic Alignment**: In the online semantic alignment process, the adversary first generates pseudo labels of the unannotated public data (row 2), and then calculates the online semantic consistency loss (row 3). The pseudo labels are also involved in sampling generator inputs from $\mathcal{D}_{pub}$ (row 6).
3. **Query Generation**: Here, the adversary uses the aggregated query generator to construct a set of the query inputs $\{\boldsymbol{x}_{query} = G(\boldsymbol{x}_{pub,1}, ..., \boldsymbol{x}_{pub,m}) | \boldsymbol{x}_{pub,1}, ..., \boldsymbol{x}_{pub,m} \in \mathcal{D}_{pub}\}$ (row 7).

---

**Algorithm 1:** Detailed Algorithm for SEEKER

---

**Input:** Victim $V$, substitute $S$, public dataset $\mathcal{D}_{pub}$, aggregated query generator $G$, number of iteration $n_l$ and size of iteration $n_s$, input number of the generator $m$
**Output:** Extracted substitute $S$

1 **Optimize** $S$ on $\mathcal{D}_{pub}$ with offline semantic consistency loss;
2 **Generate** pseudo labels $z_w$ for $\mathcal{D}_{pub}$;
3 **Calculate** online semantic consistency loss $\mathcal{L}_U$ based on $\mathcal{D}_{pub}$;
4 **for** $i \leftarrow 1$ *to* $n_l$ **do**
5    **for** $j \leftarrow 1$ *to* $n_s$ **do**
6       **Sample** $\boldsymbol{x}_{pub,1}, ..., \boldsymbol{x}_{pub,m}$ of different pseudo label classes;
7       $\boldsymbol{x}_j \leftarrow G(\boldsymbol{x}_{pub,1}, ..., \boldsymbol{x}_{pub,m})$;
8       $\boldsymbol{y}_j \leftarrow V(\boldsymbol{x}_j)$;
9    $\mathcal{Q}_i \leftarrow \{(x_j, y_j) | j = 1, ..., n_s\}$;
10   $\mathcal{Q} \leftarrow \mathcal{Q} \cup \mathcal{Q}_i$;
11    **repeat**
12       **for** $j \leftarrow 1$ *to* $|\mathcal{Q}|$ **do**
13          **Sample** $(\boldsymbol{x}_j, \boldsymbol{y}_j)$ from $\mathcal{Q}$ according to its weight;
14          $\mathcal{Q}_{train} \leftarrow \{(x_j, y_j) | j = 1, ..., n_s\}$;
15          **Calculate** the supervised substitute loss $\mathcal{L}_S$ based on $\mathcal{Q}_{train}$;
16       **Update** $S$ with $\mathcal{L}_{Sub} = \mathcal{L}_S + \lambda_U \mathcal{L}_U$;
17    **until** $\mathcal{L}$ *does not drop for 2 epochs*;
18   **Update** $G$ based on $S$ and $\mathcal{Q}_i$;
19 **return** $S$

---

4. **Supervised Train**: In the Supervised Training process, the adversary obtains the $i$-th query dataset $\mathcal{Q}_i = \{(\boldsymbol{x}_{query}, \boldsymbol{y}_{query} = V(\boldsymbol{x}_{query}))\}$ by querying the victim (row 8-9). Then the adversary updates the overall query dataset $\mathcal{Q} = \bigcup_{k=1}^{i} \mathcal{Q}_k$ (row 10). The adversary calculates the supervised loss based on $Q$ (row 12-15), and optimizes $S$ with the supervised loss and online semantic consistency loss (row 16). After substitute training, the query generator $G$ is optimized based on $S$ and $\mathcal{Q}_i$ (row 18).

## C   THEORETICAL JUSTIFICATION FOR SEMANTIC ALIGNMENT

The key insight of our semantic alignment approach is that the adversary can optimize a substitute model $S$ on a public dataset $\mathcal{D}_{pub}$ to learn features that are similar to the victim model $V$, which is trained on a secret dataset $\mathcal{D}_{secret}$. Without loss of generality, we assume the substitute model $S$ is composed of an encoding function $f$ and a fully connected layer $W$. In such a case, our observation can be reformulated as follows: if $f$ has a low semantic consistency loss and $W$ is trained on $\mathcal{D}_{secret}$ to evaluate the classification performance of $f$, $S$ has a low average classification loss on $\mathcal{D}_{secret}$. To prove this observation, we first formally define the notations.

The encoding function of $S$ belongs to $\mathcal{F}$, a class of representation functions $f : \mathcal{X} \rightarrow \mathbb{R}^d$, such that $||f(\cdot)|| \leq R$ for some $R > 0$. We denote the set of all classes in $\mathcal{D}_{pub}$ as $\mathcal{C}$, and each $c \in \mathcal{C}$ follows a probability distribution $\mathcal{D}_c$. The supervised classification loss of $S$ on $\mathcal{D}_{secret}$ can be defined as

$$\mathcal{L}_{\sup}(S) := \mathbb{E}_{(x,c) \in \mathcal{D}_{secret}} \left[ l\left( S(x)_c - S(x)_{c' \neq c} \right) \right], \tag{6}$$

where $l$ is a standard hinge loss or logistic loss, and $S = Wf$. When evaluating $S$, the best $W$ can be found by fixing $f$ and finetuning $W$. Therefore, we only denote the supervised loss of $f$ on $\mathcal{D}_{secret}$ as:

$$\mathcal{L}_{\sup}(f) = \inf_{W} \mathcal{L}_{\sup}(Wf). \tag{7}$$

Also, our offline semantic consistency loss can be formalized as

$$\mathcal{L}_C = -\mathbb{E}_{x \in \mathcal{D}_{pub}} \left[ \log \frac{\text{sim}(S(\text{aug}_M^1(x)), S(\text{aug}_M^2(x)))}{\sum_{x' \in \mathcal{D}_{pub}, x' \neq x} \text{sim}(S(x), S(x'))} \right], \tag{8}$$

Table A2: Accuracy, fidelity, and ASR of two aggregation designs.

| Encoder Architecture | Acc (%) | Fid (%) | ASR (%) |
|---|---|---|---|
| Different encoders | 88.01 | 88.94 | 96.43 |
| Base encoder+Merge encoder | 88.36 | 89.73 | 98.40 |

The loss term can be simplified as

$$\mathcal{L}_C = \frac{1}{M} \sum_{i=1}^{M} l\left(f(x_j)^T (f(x_j^+) - f(x_j'))\right), \tag{9}$$

Here, $x_j$ and $x_j^+$ are semantically equivalent data constructed by augmentations. With the notations above, we formalize a proposition as follows.

**Proposition 1.** For a substitute model $S$ composed of an encoding function $f$ and a fully connected classification layer $W$, $S$ has a low average linear classification loss on $\mathcal{D}_{secret}$ if $f$ has a low offline semantic consistency loss on $\mathcal{D}_{pub}$.

We use a theorem proposed by Saunshi *et al.* (Saunshi et al., 2019) (Theorem 4.1) to prove this proposition. Let $\mathcal{S} = \{x_j, x_j^+, x_j'\}_{j=1}^{M}$ be the triplets sampled from $\mathcal{D}_{pub}$ to optimize semantic consistency loss, $f_{|\mathcal{S}} = \left(f_t(x_j), f_t(x_j^+), f_t(x_j')\right)_{j \in [M], t \in [d]} \in \mathbb{R}^{3dM}$ be the restriction for $\mathcal{S}$ for any $f \in \mathcal{F}$, and we have a complexity measure with the following Rademacher average

$$\mathcal{R}(\mathcal{F}) = \mathbb{E}_{\sigma \in \{\pm 1\}^{3dM}} \left[ \sup_{f \in \mathcal{F}} <\sigma, f_{|\mathcal{S}}> \right]. \tag{10}$$

Let $\tau = \mathbb{E}_{c,c' \sim \rho^2}\{c = c'\}$, and we have the following theorem(Saunshi et al., 2019):

**Theorem 1.** *With probability at least* $1 - \tau$, *for all* $f \in F$

$$\mathcal{L}_{\sup}(\widehat{f}) \leq \frac{1}{(1-\tau)}\mathcal{L}_C(f) - \frac{\tau}{(1-\tau)} + \frac{1}{(1-\tau)}Gen_M \tag{11}$$

*where*

$$Gen_M = O\left(R\frac{R_s(F)}{M} + R^2\sqrt{\frac{\log\frac{1}{d}}{M}}\right). \tag{12}$$

Here, we have $Gen_M \to 0$ as $M \to \infty$, and when $\rho$ is uniform and the number of classes $|C| \to \infty$, we have that $\frac{1}{(1-\tau)} \to 0, -\frac{\tau}{(1-\tau)} \to 0$. Therefore, when the number of sampled training triplets is large and $f$ has a low offline semantic consistency loss on $\mathcal{D}_{pub}$, then $S$ has a low average linear classification loss on $\mathcal{D}_{secret}$.

## D  WEIGHT SHARING OF AGGREGATED ENCODERS

Here, we provide an alternative design of the aggregated architecture. Specifically, we have designed an architecture that both identifies the difference in the encoding of the base image and ensures the permutational invariance of the other images. Specifically, we design a base encoder for encoding the base image and a shared merge encoder to encode the other images. As shown in Table A2, this design performs slightly better than the design with different encoders for each public data.

## E  IMPLEMENTATION DETAILS FOR SEEKER

### E.1  SUBSTITUTE TRAINING SCHEME

For better reproducibility, we list some details for the substitute training process of our proposed framework as follows.

**Optimizer**: The substitute is optimized with Adam optimizer with a learning rate of 0.0003. Under a large query budget, we utilize SGD optimizer with a learning rate of 0.1, a decay rate of 0.0005, and a momentum of 0.9. Additionally, we employ a cosine annealed scheduler to gradually decay the learning rate over the epochs.

**Data augmentation**: We apply random crop and random horizontal flip to augment the public datasets, and apply random horizontal flip to augment the query dataset. For a fair comparison, the same augmentations are performed for all the other model extraction attacks.

### E.2 AGGREGATED QUERY GENERATOR

**Network architecture.** In the online stage of SEEKER, we propose aggregated query generator for informative query generation. We adopt the design of SNGAN (Miyato et al., 2018) for the aggregated query generator and employ spectral normalization to stabilize the training process of the generator.

**Optimization.** The aggregated query generator is optimized with Adam optimizer with a learning rate of 0.001. In each loop, we optimize $\mathcal{L}_{Gen}$ for 5 epochs, and each epoch is followed by 2 training epochs of optimizing $\mathcal{L}_D$.

**Loss functions.** We point out that optimizing the inconsistency loss requires querying $V$ every time $G$ is updated, which however consumes a large query budget. Instead, we only query the victim once for each public data combination $\boldsymbol{x}_{pub,1}, \boldsymbol{x}_{pub,2}, ..., \boldsymbol{x}_{pub,m}$ in a query iteration, and use the victim output to optimize the loss in this iteration. As mentioned in the main manuscript, we construct the diversity dataset $\mathcal{D}_{div}$ for optimizing the diversity loss of the aggregated query generator. We set the diversity dataset threshold $T_{div}$ to be 0.6.

### E.3 AUGMENTATIONS IN UNSUPERVISED TRAINING

Three levels of augmentations are involved in our proposed semantic alignment scheme: weak and strong augmentations of online semantic consistency loss, and medium-level augmentation for offline semantic consistency loss. Similar to Fixmatch (Sohn et al., 2020), we adopt a simple combination of random horizontal flip and random vertical flip as weak augmentation and RandAugment (Cubuk et al., 2020) as strong augmentation. Medium-level augmentation is implemented as a combination of random crop, random horizontal flip, and random color distortion, as suggested by SimCLR (Chen et al., 2020).

### E.4 WHITE-BOX ADVERSARIAL ATTACK

For all the model extraction attacks including the attack proposed by Mosafi *et al.*(Mosafi et al., 2019), Knockoff Nets (Orekondy et al., 2019), ActiveThief (Pal et al., 2020), Black-Box Ripper (Barbalau et al., 2020), DFMS-SL (Sanyal et al., 2022) and SEEKER, we use MI-FGSM (Dong et al., 2018) to perturb clean images based on the substitute models. In our experiments, we adopt the implementation of MI-FGSM from advertorch (Ding et al., 2019). As mentioned in the main manuscipt, we employ the same set of parameters for MI-FGSM across all the model extraction attacks in each experimental configuration for fair comparisons. When attacking the victim model trained on CIFAR-10, we set the perturbation level $\epsilon = 8/255$, the number of iteration steps $n_{step} = 30$. We note that the victim model trained on CIFAR-100 is more vulnerable to adversarial attacks. Therefore, we choose a smaller perturbation for the attack, where the perturbation level $\epsilon = 5/255$, the number of iteration steps $n_{step} = 30$.

### E.5 COMPUTING INFRASTRUCTURE

In our experiments, we used AMD EPYC 74F3 CPU and NVIDIA A100 Tensor Core GPU. The experiments were conducted on Ubuntu 18.04. We used Cuda 11.0.3 (NVIDIA et al., 2020), Numpy 1.21.2 (Harris et al., 2020), PyTorch 1.7.1 (Paszke et al., 2019), and Torchvision 0.8.2 (Marcel & Rodriguez, 2010) to implement our proposed framework.

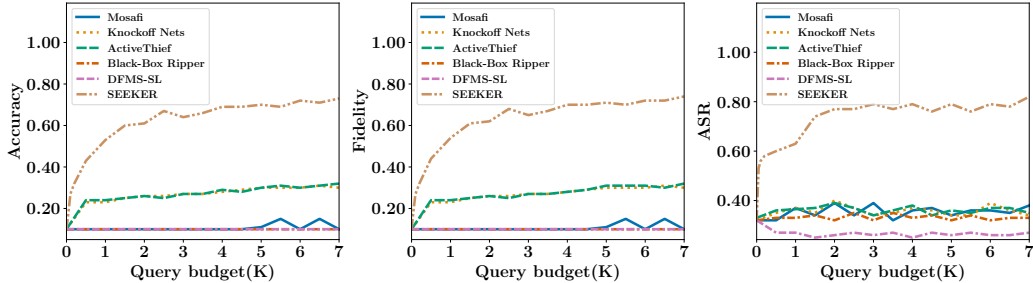

Figure A1: Accuracy, fidelity, and ASR comparisons between SEEKER and the state-of-the-art (SOTA) model extraction attacks under a query budget of 7K.

## F IMPLEMENTATION DETAILS FOR EXISTING METHODS

### F.1 MODEL EXTRACTION ATTACKS

Here, we provide implementation details of five existing model extraction attacks mentioned above: the attack proposed by Mosafi *et al.*, Knockoff Nets, ActiveThief, Black-Box Ripper, and DFMS-SL.

1) **Mosafi *et al.*** We reproduce the model extraction attack proposed by Mosafi *et al.* with PyTorch 1.7.1 (Paszke et al., 2019). While Mosafi *et al.* attack is proposed for the hard-label setting, it can be easily extended to the soft-label setting. For a fair comparison, we perform their proposed attack under the soft-label setting, using the KL Divergence loss for substitute training. The original setting requires 1 million queries for every epoch of substitute training, which largely exceeds the query budget (100K) in our experimental setting. Therefore, we run the attack for 40 epochs, and each epoch of the attack consumes 2,500 queries.

2) **Knockoff Nets.** We adopt the official implementation for Knockoff Nets in our experiments. In particular, we use the random policy in Knockoff Nets (denoted as Knockoff Nets-Random) for comparisons, since the adaptive policy assumes a different threat model from ours, and is not open-source as well.

3) **ActiveThief.** ActiveThief introduces five strategies for sampling query inputs from the public dataset: random, uncertainty, DFAL, K-center, and DFAL+K-center. We observe similar performance for the five strategies and demonstrate the results of the best-performing strategy in our experiments.

4) **Black-Box Ripper.** We use the official implementation for Black-Box Ripper for comparison. The original code uses a great number of queries to generate training samples, and we stop the querying process when the query number exceeds our query budget.

5) **DFMS-SL.** While DFMS (Sanyal et al., 2022) can be performed with or without a public dataset, we choose the former to compare with since it has a similar setting to our method. We also adopt the soft-label variant of DFMS in the comparisons.

### F.2 QUERY-BASED ADVERSARIAL ATTACKS

We compare the QER between SEEKER and three query-based black-box adversarial attacks: NES, Bandits, and Simulator attack. We calculate the QER of those methods with the open-source implementation of Ma *et al.* (Ma et al., 2021), and their detailed configurations are provided as follows. For each attack, we select a set of parameters that allows the attack to achieve the same level of ASR and attack strength as our attack. The attack strength is measured by L2 distance between the original and perturbed images.

1) **NES.** We use $l_\infty$ norm NES attack, where $\epsilon = 0.12$, the sampling variance is 0.01, the minimum learning rate is $5 \times 10^{-5}$, and the maximum learning rate is 0.05.

2) **Bandits.** We choose $l_\infty$ norm attacks in our experiments, where $\epsilon = 0.3$, the image learning rate is 0.03, the online learning rate is 1.0, and the exploration rate is 0.3.

3) **Simulator Attack.** We use the pre-trained models for launching the Simulator attack. We choose $l_\infty$ norm distance for Simulator attack. When attacking the model trained on CIFAR-10, the image learning rate is 0.004, the online learning rate is 1.0, the exploration rate is 0.3, and $\epsilon = 0.5$. When

Table A3: Model architecture comparisons. Here, we use CIFAR-10 as $\mathcal{D}_{secret}$ and CIFAR-100 as $\mathcal{D}_{pub}$.

| Model | ResNeXt | MobileNet | DenseNet | PyramidNet | ResNet-34 | WRN-28 |
|---|---|---|---|---|---|---|
| Acc (%) | 71.6 | 69.9 | 87.3 | 87.4 | 88.0 | 88.6 |
| ASR (%) | 63.5 | 56.8 | 84.7 | 91.1 | 96.4 | 96.7 |
| Params (M) | 0.68 | 3.2 | 1.0 | 9.9 | 21.3 | 36.5 |
| FLOPs (M) | 5.37 | 12.0 | 364.9 | 1466.1 | 1163.5 | 5252.6 |

Table A4: The accuracy and ASR of model extraction attacks with CIFAR-10 as $\mathcal{D}_{secret}$ certain classes of CIFAR-100 as $\mathcal{D}_{pub}$.

| Attack | CIFAR-100 (30C) | | CIFAR-100 (50C) | | CIFAR-100 (Full) | |
|---|---|---|---|---|---|---|
| | Acc (%) | ASR (%) | Acc (%) | ASR (%) | Acc (%) | ASR (%) |
| Knockoff Nets (Orekondy et al., 2019) | 44.4 | 38.4 | 48.5 | 41.5 | 75.7 | 46.9 |
| SEEKER (ours) | **66.7** | **80.8** | **78.8** | **89.2** | **88.0** | **96.4** |

attacking the model trained on CIFAR-100, the image learning rate is 0.02, the online learning rate is 1.0, the exploration rate is 0.3, and $\epsilon = 1.5$.

## G  COMPARISONS UNDER SMALLER QUERY BUDGETS

In Figure 5 of our main manuscript, we provide the accuracy, fidelity, and ASR of the SOTA model extraction attacks across different query budgets ranging from 500 to 0.5M. To better demonstrate the query-efficiency of different attacks, we further illustrate the accuracy, fidelity, and ASR of the attacks across a range of query budgets from 0 to 7K. As shown in Figure A1, SEEKER outperforms existing attacks by a large margin within 7K queries in terms of accuracy, fidelity, and ASR. We also point out that the attack proposed by Mosafi *et al.*, Black-box Ripper, and DFMS-SL only attain limited performance under an extremely small query budget. In contrast, our attack is able to extract a substitute with an accuracy of 73.3% and an ASR of 72.5% with 7K queries.

## H  COMPARISONS WITH QUERY-BASED ATTACKS

Figure A2 demonstrates the QER comparisons between our attack and query-based attacks when using CIFAR-100 as $\mathcal{D}_{secret}$. Similar to the results in the main manuscript, we observe that query-based attacks are more query-efficient than our method only when producing a small number of adversarial samples. However, the QER of query-based attacks is easily outperformed by our attack when more than 5K adversarial samples are required. Moreover, substitute-based attacks (e.g. SEEKER) attain asymptotically higher query-efficiency than query-based attacks. For example, the black-box adversarial attack based on our approach achieves $3\times$ QER over that of Simulator attack when crafting 15K adversarial samples.

## I  IMPACT OF SUBSTITUTE MODEL ARCHITECTURES

To evaluate how different substitute model architectures affect the performance of our method, we test six neural network architectures for the substitute model: ResNeXt, MobileNet, DenseNet, PyramidNet, ResNet-34, and WRN-28. Here, we use ResNet-34 as the model architecture of the victim model. As shown in Table A3, model architectures with more parameters and FLOPs tend to achieve better extraction performance. This observation demonstrates that the performance differences across model architectures are mainly due to the inherent architectural design of the substitute neural network model. We note the similarities between the architectures of the substitute model and the victim model can also have small but observable impacts on extraction performance. For instance, ResNeXt attains slightly better performance than MobileNet, even though the latter has more parameters and FLOPs.

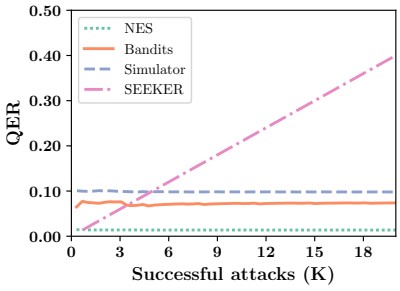

Figure A2: Query efficiency comparisons between our method and query-based attacks with CIFAR-100 as $\mathcal{D}_{secret}$ and CIFAR-10 as $\mathcal{D}_{pub}$.

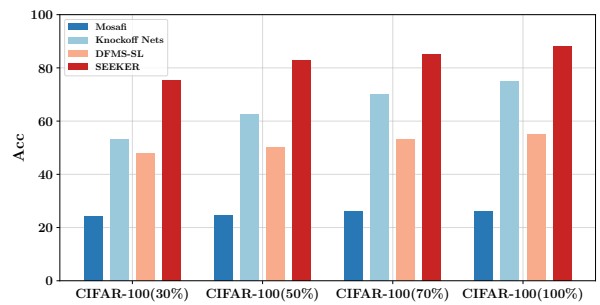

Figure A3: The accuracy of different model extraction attacks with CIFAR-10 as $\mathcal{D}_{secret}$ and randomly sampled subsets of CIFAR-100 as $\mathcal{D}_{pub}$.

Table A5: Accuracy, fidelity, and ASR of different model extraction attacks with different public datasets. Here, we use CIFAR-10 as $\mathcal{D}_{secret}$.

| $\mathcal{D}_{pub}$ | Attack | Acc (%) | Fid (%) | ASR (%) |
|---|---|---|---|---|
| Caltech256 (Training set) | Knockoff Nets | 50.05 | 50.75 | 42.63 |
| | SEEKER (ours) | **65.71** | **66.54** | **78.26** |
| STL10 (Unlabeled set) | Knockoff Nets | 78.74 | 79.84 | 52.36 |
| | SEEKER (ours) | **88.57** | **89.89** | **98.97** |

## J    IMPACT OF DIFFERENT PUBLIC DATASETS

To further demonstrate the generalization capability of our attack, we have compared our method with the best-performing attack, Knockoff Nets, on different public datasets. Here, we use CIFAR-10 as $\mathcal{D}_{secret}$ and use the training set of Caltech256 and unlabeled set of STL10 as $\mathcal{D}_{pub}$. As shown in Table A5, we have found that our method generalizes significantly better than the state-of-the-art attack on different datasets.

## K    IMPACT OF PUBLIC DATASET SIZE

For a certain secret dataset $\mathcal{D}_{secret}$, we sample subsets from the public dataset $\mathcal{D}_{pub}$ to analyze how the size of $\mathcal{D}_{pub}$ impacts the performance of the model extraction attacks. To this end, we design two experiments according to the sampling strategy of $\mathcal{D}_{pub}$. In the first experiment, we select all of the data in certain classes of CIFAR-100 as $\mathcal{D}_{pub}$. In the second experiment, we randomly sample subsets of CIFAR-100 as $\mathcal{D}_{pub}$ with equal sampling probabilities for each class. For both experiments, we use CIFAR-10 as $\mathcal{D}_{secret}$ to train the victim model.

In the first experiment, we compare the accuracy and ASR between SEEKER and Knockoff Nets (the best-performing method within 100K query budget) with 30 classes of CIFAR-100, 50 classes of CIFAR-100, and the full CIFAR-100 dataset as $\mathcal{D}_{pub}$. As shown in Table A4, our attack can extract a substitute model more effectively than Knockoff Nets when the adversary can only access a small number of classes of $\mathcal{D}_{pub}$. For instance, SEEKER achieves 66.7% accuracy and 80.8% ASR even if only 30 classes of CIFAR-100 are publicly available, whereas Knockoff Nets only attains 44.4% accuracy and 38.4% ASR under the same condition.

In the second experiment, we randomly sample 30%, 50%, 70%, and 100% of CIFAR-100 uniformly across the labels as $\mathcal{D}_{pub}$ to perform different model extraction attacks. Here, we note that 30 classes of CIFAR-100 in the first experiment and randomly sampled 30% subset of CIFAR-100 in the second experiment are both 30% data from the CIFAR-100 dataset but produced with different sampling strategies. The results in Figure A3 demonstrate that our method achieves higher accuracy than the other SOTA attacks across different public dataset sizes. We also point out that the accuracy of our attack does not drop significantly as the size of $\mathcal{D}_{pub}$ decreases. The results of both experiments reflect that SEEKER is more effective in leveraging the information in the public datasets.

Table A6: Ablation experiment for training dataset during the substitute training stage

| Training dataset | Acc (%) | Fid (%) | ASR (%) |
|---|---|---|---|
| New queries | 83.8 | 85.2 | 93.2 |
| All queries | 86.9 | 88.2 | 95.8 |
| Sampled queries (ours) | **88.0** | **88.9** | **96.4** |

Table A7: Ablation experiment for loss-based training termination

| Training condition | | Acc (%) | Fid (%) | ASR (%) |
|---|---|---|---|---|
| Number of epochs per iteration | 20 epochs | 82.6 | 83.8 | 91.1 |
| | 40 epochs | 86.8 | 88.2 | 95.9 |
| | 60 epochs | 87.4 | 88.8 | **96.9** |
| | 80 epochs | 81.8 | 83.0 | 94.0 |
| | 100 epochs | 82.3 | 83.5 | 86.2 |
| Automatic (ours) | | **88.0** | **88.9** | 96.4 |

## L  ABLATION STUDIES

We designed a set of ablation experiments to examine to study how much each component in the framework contribute to the overall results. Our main manuscript introduces the results for augmentation invariant unsupervised training and aggregated query generator. Here, we present the results for two remedies for preventing substitute overfitting: weighted sampling and loss-based training termination.

To evaluate the effect of weighted sampling, we compare the accuracy, fidelity, and attack success rate (ASR) of substitute models trained with different datasets, including complete query dataset, new query dataset, and weighted sampled dataset. As shown in Table A6, weighted sampling achieves higher accuracy and ASR than the complete query dataset and new query dataset. For example, the weighted sampled dataset achieves 1.1% higher accuracy than the complete query dataset and 3.2% higher ASR than the new query dataset. In the above experiments, we use a weight factor $\alpha = 0.8$ for the weighted sampling strategy.

In addition to weighted sampling, we study the contribution of loss-based termination by comparing the performance of $S$ trained under a manually controlled condition and an automatic condition. In the manually controlled conditions, we set a fixed number of training epochs (20, 40, 60, 80, 100) for each loop. In the automatic condition, substitute training terminates if the loss value does not decrease for two consecutive epochs. As shown in Table A7, the substitute model trained with loss-based termination achieves similar or better accuracy, fidelity, and ASR when compared to the models trained in the manually controlled condition.

