# OpenReview forum: "SEEKER: Semi-Supervised Knowledge Transfer for Query-Efficient Model Extraction"
_ICLR.cc/2024/Conference — Submitted to ICLR 2024_

### Official Review · Reviewer_8x4q · 2023-10-30

**Soundness:** 3 good
**Presentation:** 1 poor
**Contribution:** 3 good
**Rating:** 5
**Confidence:** 2

**Summary:**

The paper addresses the vulnerability of Deep Neural Networks (DNNs) to model extraction attacks, even when the models are accessed in a black-box manner. These attacks allow adversaries to create a substitute model that mimics the original model by querying the black-box model with unlabeled inputs. The paper introduces SEEKER, a query-efficient model extraction framework that leverages semi-supervised public knowledge transfer. The framework incorporates an offline stage for pre-training the substitute model without any query costs, a semantic alignment scheme, and a multi-encoder query generator. Experimental results indicate that SEEKER significantly improves query efficiency while maintaining high accuracy and attack success rate (ASR) compared to state-of-the-art methods.

**Strengths:**

1. SEEKER introduces a novel approach to model extraction that combines offline pre-training with semantic alignment, reducing the need for extensive querying. The paper claims that SEEKER can reduce the query budget by over 50 times compared to existing methods while achieving comparable or better accuracy.
2. The experimental results are thorough and clearly presented.

**Weaknesses:**

I like the technical merit in this paper. My major concerns are around the presentation that might require significant modification of the main body. I'm willing to raise the score if the following concerns are addressed.
1. In the methodology part, there is no explicit discussion or intuition why the proposed method can improve the query efficiency, i.e., reduce the number of queries. If so, what is the query efficiency just a side effect? We need more justification that only experimental results.
2. The paper presents a complex methodology without providing much intuition. I can get a main idea of whole methodology. But the motivation of each part is not clear. For example, in section 3.2, why a good framework should be designed this way.

**Questions:**

Is it possible to provide a simpler graph than Figure 1 for illustrating the main idea?

---

> ### Author Response · Authors · 2023-11-20
> **Response to Reviewer 8x4q (1/2)**
>
> We thank the reviewer for acknowledging the technical merits of our manuscript and pointing out the problems in the presentation. We would like to make the following clarification on our contribution and make corresponding modifications to the manuscript.
>
> > 1. Give explicit discussion or intuition why the proposed method can improve the query efficiency.
>
> We appreciate the suggestion of the reviewer.
> Our work improves the query-efficiency of model extraction attacks by proposing a *query-free self-supervised* scheme and a *query-efficient supervised* approach.
> Both methods are aimed at extracting the knowledge from the public dataset effectively to reduce the query cost.
> We would like to discuss the motivation of each method as follows.
>
> **Query-free Self-supervised Scheme.**
> We note that the conventional public dataset based model extraction attacks only use public data for crafting queries to the victim model.
> In contrast, we develop a self-supervised training scheme to optimize the substitute model exclusively based on the public dataset, which does not require any queries to the victim model.
> Our approach builds on the assumption that a well-trained victim model
> maintains semantic consistency, i.e., outputs similar representations for different images featuring the same object.
> Using this semantic consistency as an additional prior, our method optimizes the substitute model to learn similar encoding patterns to the victim model.
> We also design different variants of the scheme according to the characteristics of offline and online stages that reduce the query cost of both stages.
>
> **Query-efficient Supervised Scheme.**
> To further improve the query-efficiency of the supervised learning stage of conventional attacks, we focus on synthesizing queries with richer information.
> We point out that most existing model extraction methods based on public data feature a one-to-one correspondence between the public data and the generated query, which limits the information density in the query.
> To increase the information diversity in the query generation process, we propose to parallelly process different public data when synthesizing a single query.
> We design a multi-encoder aggregation design and reconstruction loss to merge the information from different public data, and develop inconsistency and diversity loss to ensure such information is fully leveraged for minimizing the gap between the substitute and the victim models.
>
> In our revised manuscript, we also made major revisions to the overview of our method (Section 3.2), the semantic alignment scheme (Section 3.3), and the aggregated generation method (Section 3.4), including a more detailed discussion of how our attack reduces the query cost of the model extraction process.

---

> ### Author Response · Authors · 2023-11-20
> **Response to Reviewer 8x4q (2/2)**
>
> > 2. Explain the motivation behind each part of the framework.
>
> We appreciate the suggestion of the reviewer.
> Here, we would like to introduce how each component of the method contributes to the overall performance and how they are incorporated into the framework.
>
> Our framework comprises four main procedures: offline semantic alignment, online semantic alignment, query generation, and supervised training.
> The offline semantic alignment belongs to an independent offline pre-training stage, and the adversary iterates through the online semantic alignment, query generation, and supervised training procedures during the online stage.
> Among these procedures, conventional model extraction attacks only include query generation and supervised training procedures in the online stage, i.e., generate query inputs, query the victim model, and use the victim feedback to optimize the query generation process.
> Our framework improves the existing pipeline by 1) introducing an offline stage (offline semantic alignment), 2) designing the online semantic alignment to assist the conventional supervised training scheme, and 3) proposing a novel aggregation generator for the query generation process.
>
>
> **Offline semantic alignment.**
> In contrast to existing online attacks, we design an offline stage to pre-train the substitute model.
> During this stage, the adversary can not directly interact with the victim, but have access to a public dataset.
> Under such circumstances, we design an offline semantic alignment strategy that leverages the public dataset to optimize the encoding layers of the substitute model in a self-supervised manner.
> Notably, we identify semantic consistency as a common property in neural networks and utilize the prior to train the substitute model.
> We also design a symmetric network architecture and projection layers that are suitable for the offline stage.
>
> **Online semantic alignment.**
> In addition to the offline semantic alignment, we further develop a self-supervised learning scheme to augment the online stage.
> Different from the offline stage, the substitute model in the online stage is trained with the victim output, and has similar outputs to the victim model.
> Therefore, the online semantic alignment scheme is aimed at further improving the performance of the substitute.
> Online semantic alignment uses weak augmentations for the substitute to produce pseudo labels, and aligns the substitute output of strongly augmented data with the pseudo labels.
> This approach gains extra accuracy increase for the substitute model over the original supervised training scheme.
>
> **Informative query generation.**
> We also make an important contribution to the conventional model extraction framework by designing a query generation procedure with richer information.
> Our proposed query generation process achieves three main goals to increase the information density of synthesized queries:
> 1) *Aggregating*: the generator can effectively merge information from multiple public data,
> 2) *Informative*: the generator can produce information-extracting queries that minimize the gap between the substitute and the victim,
> 3) *Stealthy*: the synthesized queries maintain the structure of a natural image instead of collapsing into indistinguishable patterns.
> We propose
> an aggregation architecture and three loss functions for the query generator to achieve all three goals at the same time.
>
> We have also modified our manuscript to provide an overall discussion on the design of the framework (Section 3.2), and provide more motivation for each main component of our framework (Section 3.3, Section 3.4).
>
> > 3. Is it possible to provide a simpler graph than Figure 1 for illustrating the main idea?
>
> We thank the reviewer for the suggestion.
> We have modified Figure 1 to provide a clearer illustration of our two main contributions.
> Specifically, we have simplified the illustration of our proposed aggregated query generation to highlight the novel aggregation design.

---

### Official Review · Reviewer_WA7v · 2023-10-30

**Soundness:** 3 good
**Presentation:** 3 good
**Contribution:** 2 fair
**Rating:** 5
**Confidence:** 3

**Summary:**

Model extraction attacks focus on creating a substitute model whose performance resembles a victim model’s performance; this is achieved by querying the victim model with a selection of samples and observing the output behavior. Among other things, doing so allows whitebox adversarial attacks to be used to target the victim model. This particular paper proposes two major components for improving such attacks. The first is a semantic alignment, done both as offline pre-training and during querying. The second is a way of parallelizing the query generation process by encoding the information of multiple samples into a single sample. The result is a method that is much more query-efficient than prior methods, gaining high substitute model, fidelity, and attack success rate with far fewer queries, while also being strong in the large query regime as well.

**Strengths:**

## S1. Query efficiency
A central claim of this paper is that the proposed method reduces the number of queries needed to perform a successful attack. This is important as a large number of queries renders a method slow and computationally expensive, while also significantly raising the risk of being detected (and stopped) by the target system. Figure 5 effectively illustrates how much faster the substitute model’s metrics can improve using the proposed approach vs the baseline methods. Highlight claim is that this can be up to 50x more query efficient.

## S2. Empirical Results
The empirical results of this paper are generally strong. Again, Figure 5 provides an excellent summary, showing higher accuracy, fidelity, and ASR vs competing methods, with fewer number of queries. At the (very) high query range, the proposed approach also shines (Table 2), barely edging out DFMS-SL. Results also appear to generalize to other model architectures (Table 3).


## S3. Generally clearly written
The paper is clearly written and easy to understand. That said, the paper could still use another round of editing: there are a few typos, some potential improvements to the technical notation, and a number of other issues to fix (see Miscellaneous under Weaknesses). Overall though, this paper was easy to read.

**Weaknesses:**

## W1. Novelty
Semantic consistency is presented as one of the two methodological improvements proposed by the authors. The concept, however, doesn’t strike me as particularly new. Offline semantic alignment is simply self-supervised pre-training on a different dataset (very reminiscent of [b], in fact); it’s not particularly surprising that this is helpful, as that’s more or less the whole point of self-supervised learning. The similarities in Grad-CAM visualizations doesn’t necessarily have much to do with model extraction: it may be more due to having stronger features, which presumably the victim model also has, resulting in strong correlation. Online semantic consistency is more or less just knowledge distillation [d] with augmentations.

## W2. Aggregation Design
a) Why the query perturbation/residual is added to the first input sample $x_{pub,1}$ (as opposed to, say, $x_{pub,2}$) isn’t clear and seems like an arbitrary choice. It would seem like there could be a smarter way to select which of the $m$ samples to chose as the base sample. \
b) It’s also not clear to me why there is a separate encoder per query index. As far as I can tell, the relative ordering of queries within a parallel batch is not meaningful, so there doesn’t seem to be any added value behind having separate encoders, vs just having a single shared encoder or some other permutationally invariant design (e.g. a transformer).\
c) I don’t fully understand the design of the reconstruction loss. Why do we care about reconstructing the input data? We already have the input data. Rather than generating new images through an encoder and decoder, what about simpler methods like MixUp or CutMix? The formulation in Equation 3 also involves a significant number of hyperparameters $\alpha_j$ to tune, and it’s not clear why they should be meaningfully different; as in b) above, it doesn’t seem like the order of the samples have any inherent meaning, so why for example should $\alpha_2 \neq \alpha_3$?

## W3. Source of empirical improvements
The ablation study in Section 4.4 is very helpful, but it’s also somewhat concerning too. From this table, it appears that the offline semantic alignment is the bulk of the source of improvements. While doing this offline semantic alignment is reasonable, as stated in W1 above, it’s really another name for starting off with a stronger substitute model through standard self-supervised pre-training, which is already well known by the community to be generally effective for many downstream tasks. The form of self-supervised learning used by the authors is fairly standard, so it’s not clear to me if this really can be counted as the authors’ contribution.

## W4. Section 4.5
Analysis of the proposed method’s performance against defense mechanisms is welcome, but most of the actual results (particularly the paragraph on “Passive Defenses”) appear to be absent from the main paper and deferred to the Appendix. If these analyses are going to be introduced in the main paper, then at least some quantitative results should be included.

## Miscellaneous:
- pg 2: “Mosafi et al.(Mosafi et al., 2019)” <= use in-text citation
- pg 3: “with [a] generative adversarial network”
- Section 2.2: This section leaves out transfer-based blackbox attacks, which generate attacks on the attacker’s own model, which are then given to the target (victim) model. Such approaches have also been combined with query-based attacks, with very low query requirements, e.g. [a].
- pg 4: “a[n] aggregated”
- pg 4: Why not denote $x_{pub}$ as $x_p$ instead, to match $D_P$?
- pg 4: “$i$-th query dataset” <= isn’t this just a sample and label, not a dataset?
- pg 4: “that share the same weight[s]”
- pg 4: This is a standard Siamese network approach to self-supervised learning; more such methods should be cited: e.g. [b,c]
- pg 5: “we [a] propose aggregated query generator”
- Eq 5: If I’m understanding this loss correctly, I think this is for the $i+1$th iteration, not $i$th. Also, as written, this implies we only care about a new query sample not resembling the immediately previous query; this doesn’t prevent a query for example from flipping back and forth between two queries.
- pg 6: “Second, [w]e design”
- Fig 5: Caption should mention what dataset this is on. Text doesn’t mention it either.
- Table 2: What dataset is this? Text doesn’t say either.
- Table 3: What dataset is this? Text doesn’t say either.
- Table 4: What dataset is this? Text doesn’t say either.


[a] Inkawhich, Nathan, et al. "Perturbing across the feature hierarchy to improve standard and strict blackbox attack transferability." NeurIPS 2020. \
[b] Chen, Xinlei, and Kaiming He. "Exploring simple siamese representation learning." CVPR 2021.\
[c] He, Kaiming, et al. "Momentum contrast for unsupervised visual representation learning." CVPR 2020.\
[d] Hinton, Geoffrey, Oriol Vinyals, and Jeff Dean. "Distilling the knowledge in a neural network." 2015.\

**Questions:**

Q1. How does the similarity of the public dataset to the target model’s dataset affect results? The main results in Table 1 show significant overlap: CIFAR-10 is used as the public dataset when CIFAR-100 is the hidden dataset, and vice versa, and even Tiny-ImageNet has strong similarities compared to CIFAR 10/100.\
Q2. Is the substitute model the same architecture as the victim model? Is that a reasonable assumption to make? What if they’re different?

**Details Of Ethics Concerns:**

Although I answered “No” above, this paper does deal with subject matter model extraction and adversarial attacks. Obviously, these such approaches can be applied for nefarious purposes, including theft or causing systems to behave in unexpected ways, leading to negative outcomes. With that said though, I don’t believe this paper doesn’t necessarily introduce any ethical concerns beyond what is common for other works working on this problem.

---

> ### Author Response · Authors · 2023-11-20
> **Response to Reviewer WA7v (1/3)**
>
> We appreciate the reviewer for making detailed reviews and insightful comments on our work.
> In this response, we have made several clarifications for our work and made some modifications to our manuscript according to the valuable suggestions of the reviewer.
> The revised parts are marked as blue.
>
> ## W1. The novelty of semantic alignment.
>
> We thank the reviewers for the questions. We would like to make the following clarifications.
>
>
> First, we point out that no existing work in the field of model extraction has explored self-supervised learning.
> Although the Siamese design is common in self-supervised learning, we are the first to find that the design can be effective in model extraction, and adapt it with specific modules that are suitable for both the offline and online stages of the model extraction process.
> Moreover, we are the first to design an offline pre-training stage for model extraction.
> When designing the offline stage, we note that simply using *supervised* pre-training method for extracting *supervised* models has two major disadvantages: 1) supervised pre-training makes a strong and impractical assumption that the public dataset is annotated, and 2) such scheme performs worse that the proposed self-supervised scheme.
> In contrast, we demonstrate the *self-supervised* learning approach not only eliminates the need for public annotation, but also demonstrates superior performance in extracting *supervised* models, as shown in Tab. R1.
> Additionally, we would like to clarify that knowledge distillation is not applicable for online self-supervised training.
> Knowledge distillation minimizes the distance between the outputs of the teacher and student model, but the two models are the same in a self-supervised setting.
> Therefore, we design weak and strong augmentations to introduce semantic consistency as an extra prior that optimizes the substitute model in a self-supervised manner.
>
> Table R1. Comparisons between supervised pre-training and our proposed self-supervised pre-training
>
> |Pre-training strategy|Acc (\%)|Fid (\%)|ASR (\%)|
> |---|---|---|---|
> |Baseline|74.30|75.87|46.92|
> |Baseline+supervised pre-training|81.56|82.02|78.31|
> |Baseline+self-supervised pre-training (ours)|85.11|86.26|84.74|
>
> Finally, we kindly point out that we carefully design the self-supervised approach according to the characteristics of the offline and online extraction stages.
> Specifically, we would like to highlight the following contributions in designing self-supervised methods for two stages.
>
> * Network architecture:
> In the offline stage, the adversary does not have any knowledge of the victim model, so we design a symmetric architecture for this stage.
> In the online stage, as the substitute model already has similar prediction scores to the victim, we design a weakly augmented branch without backpropagation to produce pseudo labels and a strongly augmented branch to learn richer semantic information.
> We also design different levels of augmentations accordingly for each stage.
>
> * Feature space for semantic alignment:
> In the offline stage, we perform semantic alignment in a high-dimensional feature space to transfer representational information from the public dataset.
> In the online stage, we directly optimize semantic consistency loss on the classification scores, as the substitute predictions are relatively accurate in this stage.

---

> ### Author Response · Authors · 2023-11-20
> **Response to Reviewer WA7v (2/3)**
>
> ## W2 Concerns regarding the designs of aggregated query generator
> > a) Is there a way to select the best base image out of $m$ images instead of using $x_{pub, 1}?$
>
> We thank the reviewer for the valuable suggestion.
> We would like to point out that selecting the most suitable public data as the base image requires comparing the $m$ images based on the return of $V$, which causes extra queries and does not align with our goal of query efficiency.
> In such a case, there is an exploration-utilization tradeoff between high query-efficiency and finding the theoretically optimal base image.
> Therefore, we choose to achieve higher query-efficiency by randomly selecting a base image.
>
> > b) Why not use encoders with shared weights in the aggregated design?
>
> This is a very insightful comment, and we have also considered the shared encoder design when designing the aggregated query generator.
> As shown in Tab. R2, we have found that using a shared encoder performs slightly worse than using different encoders for different indexes, and finally, we chose the latter as the current design.
>
> Table R2. Comparisons between Encoder Architecture
>
> |Encoder Architecture|Acc (\%)|Fid (\%)|ASR (\%)|
> |---|---|---|---|
> |shared encoder|86.72|88.26|94.82|
> |different encoders (ours)|88.01|88.94|96.43|
>
> A potential explanation for this observation is that the encoders for the base image (i.e., $x_{pub,1}$) and the merged images  (i.e., $x_{pub,2}, ..., x_{pub,m}$) learn different parameters for encoding different inputs.
>
>
> > c) Explain the reconstruction loss: why use it, how to tune $\alpha_j$, and if simple aggregation operations work.
>
> We design the reconstruction loss that optimizes the query generator to effectively merge the information from different images, and we introduce $\alpha_j$ to ensure the generated image largely resembles the base image instead of collapsing into unnatural patterns.
> Therefore, we set the $\alpha_1$ to be larger for the base image and $\alpha_2,...,\alpha_m$ to be smaller for the other images.
> Empirically, we find setting $\alpha_2=...=\alpha_m$ achieves a good performance, and did not carefully tune $\alpha$ for each image.
>
> As for simple aggregations,
> we have considered a few basic aggregation operations when choosing the aggregation design.
> We found that such aggregations have two shortcomings: 1) such methods produce unnatural query images that can be easily detected by the victim, and 2) such methods provide no significant improvements or even worse results when compared to the baseline.
> As shown in Tab. R3, we have found our proposed aggregated generator performs significantly better than simple linear combinations, such as average, MixUp (weighted average with a random probability), or CutMix.
>
> Table R3. Comparisons among different aggregation strategies
> |Aggregation Strategy|Acc (\%)|Fid (\%)|ASR (\%)|
> |---|---|---|---|
> |Baseline|74.30|75.87|46.92|
> |Average|72.77|73.65|56.32|
> |MixUp|73.79|74.37|62.83|
> |CutMix|75.41|76.49|64.47|
> |Aggregated generator (ours)|84.75|86.23|94.33|
>
>
> ## W3. Source of empirical improvements
>
> We demonstrate that simply using supervised pre-training is much less effective than our offline self-supervised pre-training.
> More importantly, we develop offline and online self-supervised training schemes according to the characteristics of the two stages, and the combination of both methods results in a strong overall performance.
> Finally, we note that our proposed aggregated generator not only contributes to the overall accuracy at a comparable level when compared to offline semantic alignment, but also makes a more significant boost in ASR (11.69\%+) than offline semantic alignment.
>
>
> ## W4 Section 4.5
> We thank the reviewer for the suggestion, and we move more detailed quantitative results of the appendix to the main manuscript.
> The modified content is marked as blue in paragraph 3, Section 4.5 in the revised paper.

---

> ### Author Response · Authors · 2023-11-20
> **Response to Reviewer WA7v (3/3)**
>
> ## Miscellaneous
> We thank the reviewer for checking our manuscript in detail and proposing issues for us to further improve the quality of the manuscript.
> We have made the following modifications:
>
> * Section 2.1, used in-time citation: "Mosafi et al. (2019)"
> * Fixed the typo in Section 2.2: "with [a] generative adversarial network"
> * We have added the following description regarding transfer-based attacks in Section 2.2: "Generally speaking, we can classify black-box adversarial attacks into two categories:
> substitute-based, transfer-based, and query-based.
> First,
> a number of substitute-based attacks have already demonstrated the effectiveness
> of using the extracted substitute model as a base for launching black-box
> adversarial attacks.
> Additionally, transfer-based attacks assume the adversary can obtain a substitute model trained on the same dataset as the victim model, and focus on improving the transferability of the adversarial samples synthesized based on the substitute model.
> Therefore, substitute-based and transfer-based attacks are generally complementary to each other."
> * Fixed the typo in Section 3.2: "a[n] aggregated"
> * We denote the public dataset as $\mathcal{D}\_{pub}$ to match $x_{pub}$, which is clearer to read. We also denote the secret dataset as $\mathcal{D}\_{secret}$ in our revised manuscript.
> * We use a new index $j$ for a more clear description in Section 3.2: "Here, we take the $i$-th iteration with the query number of $n_i$ as
> an example...the adversary obtains the $i$-th query dataset $\mathcal{Q}\_i=\{(x\_{query}^{i,j}, y\_{query}^{i,j} = V(x\_{query}^{i,j}) | j = 1, ..., n_i\}$ by querying the victim.""
> * Fixed the typo in Section 3.3.1: "that share the same weight[s]"
> * We have cited more papers for Siamese network architecture in Section 3.3.1.
> * Fixed the typo in Section 3.4: "we propose [an] aggregated query generator"
> * We have changed the notation $x_{query}^i$ to $x_{query}$. In the diversity loss, we are aimed at reducing the distance between the queries in the $i$-th iteration and all past queries. We thank the reviewer for pointing out the typo in the index.
> * Fixed the typo in Section 3.5: "Second, [w]e design"
> * Denoted the experimental setting for Fig.5, Tab.2, Tab.3, and Tab.4 in Section 4.1:  "In Figure 5, Table 2,
> Table 3 and Table 4, we use CIFAR-10 as $\mathcal{D}\_{secret}$ and CIFAR-100 as $\mathcal{D}\_{pub}$ "
>
> ## Q1. How does the similarity of the public dataset to the victim model’s dataset affect results?
> We thank the reviewer for the question. We would like to point out that our attacks can still achieve state-of-the-art performance when the two datasets are very dissimilar in distribution.
> As shown in Tab. 1 of the main manuscript, we have tested the performance of different attacks when the secret dataset is CIFAR-100, and the public dataset is ImageNet.
> The CIFAR-100 and ImageNet datasets are collected independently, have distinctive classes without overlap, and also have different resolutions.
> With such different distributions between the two datasets, our attack is able to demonstrate significantly stronger performance than the existing attacks.
>
> ## Q2. What if the substitute model has a different architecture from the victim model?
> We kindly point out that our attack obtains better accuracy and ASR than the best-performing attack across different substitute model architectures, as shown in Tab. 3 of the main manuscript.
> Moreover, we have compared the performance of our attack across more substitute model architectures in Tab. A2 of our appendix.

---

> ### Comment · Reviewer_WA7v · 2023-11-22
>
> I thank the authors for their responses, as well as the updates they've made to the paper. I've also read the other reviewers comments.
>
> Despite the authors' arguments, I'm still not entirely convinced by the novelty here. The field as a whole is well-aware at this point that self-supervised learning is effective, and I don't believe that "Self-supervised Learning Applied to X" for every problem setting is necessary or novel. I understand that there are some nuances to the method beyond that, but I find these modifications relatively small. While novelty is not the sole basis of my score, it's something that I consider as part of my holistic evaluation.
>
> I also remain concerned about the methodology. I strongly believe we should not add ordering dependencies into a method where there are none. I accept that choosing a base image without increasing query count is nontrivial (although perhaps some comparison between images offline without querying the victim may be possible). However, the use of separate encoders is not something that makes sense to me, as again, the order of the images here is completely random and does not contain any learnable patterns. Perhaps there's some tangible benefit from having a different encoder for the base image given its distinct role compared to the others, but the method could still benefit from some re-thinking to make it more permutationally invariant.
>
> Q1: As I stated in my original question, while yes, there are some differences, CIFAR and TinyImageNet are both object-centric RGB datasets, and features will tend to transfer relatively well between them. There are much more different image datasets out there.
>
> Q2: For the different architectures in Table 3 and A2, what is the victim model architecture?

---

> ### Author Response · Authors · 2023-11-23
> **Response to Reviewer WA7v**
>
> >  I don't believe that applying "Self-supervised Learning Applied to X" for every problem setting is necessary or novel. I understand that there are some nuances to the method beyond that, but I find these modifications relatively small.
>
> We thank the question of the reviewer regarding the novelty of self-supervised model extraction.
> We would like to clarify that it is necessary and novel to design self-supervised learning approach for model extraction.
> As most existing model extraction attacks only focus on information-extracting **data synthesis**, we point out a different approach regarding the **training procedure** of the substitute model.
> Our self-supervised approach is orthogonal to existing studies on supervised model extraction, and our proposed offline and online self-supervised methods can be combined with existing supervised learning attacks for significantly improved accuracy, ASR, and query-efficiency without extra query cost.
> We believe this finding is impactful on the topic of model extraction, uncovering a different research direction for this field.
> Furthermore, by proposing the self-supervised model extraction, we reveal an important finding that the utilization of public data can pose serious security threats to private training data and confidential models, raising concerns about the protection of AI privacy and robustness.
>
> Additionally, we kindly point out that the modifications we made for the original self-supervised framework are novel and effective for model extraction.
> Particularly, we have designed the stage-specific architecture, the feature space for alignment, and different semantically equivalent branches, all of which contribute to the strong overall performance.
>
> > Although there's some tangible benefit from having a different encoder for the base image given its distinct role compared to the others, the method could still benefit from some re-thinking to make it more permutationally invariant.
>
> We thank the reviewer for the suggestion.
> We have designed another architecture for the aggregated generator that both identifies the difference in the encoding of the base image and ensures the permutational invariance of the other images. Specifically, we design a *base encoder* for encoding the base image and a shared *merge encoder* to encode the other images. Also, we adopt a new formulation of the reconstruction loss with only two weight hyperparameters, one for the base image and one for the other merged images:
> $$\mathcal{L}\_R = \frac{1}{m}(\alpha_1||G(x\_{pub,1}, x\_{pub,2}, ..., x\_{pub,m}) -  x\_{pub,1}||\_2+\sum_{j=2}^m\alpha_2||G(x\_{pub,1}, x\_{pub,2}, ..., x\_{pub,m}) -  x\_{pub,j}||\_2),$$
>
> Here, we assign $\alpha_1>\alpha_2$ to ensure that the generated query preserves the structure of the base image.
> This new formulation explicitly constrains the permutational invariance of the merged images and also gives a higher weight for the base image.
> The above design ensures that the structure of the base image is better preserved while encoding the other images in a permutationally invariant manner.
>
> Table R1 shows this design obtains slightly better performance than the original design with different encoders. We have also included this design and results in Section D of our revised appendix and the new formulation of reconstruction loss in Section 3.4.2 of our revised manuscript.
>
> Table R1. Accuracy, fidelity, and ASR of two aggregation designs
>
> |Encoder Architecture|Acc (\%)|Fid (\%)|ASR (\%)|
> |---|---|---|---|
> |different encoders|88.01|88.94|96.43|
> |Base encoder+Merge encoder|88.36|89.73|98.40|
>
> > Q1: Provide the results for more different public datasets.
>
> We thank the suggestion of the reviewer. We have compared our method with the best-performing attack, Knockoff Nets, on different public datasets.
> As shown in Table R2,
> we have found that our method generalizes significantly better than the state-of-the-art attack on different datasets.
> Here, the secret dataset is CIFAR-10.
> We have also included the results in Section J of our revised manuscript.
> Additionally, we are running experiments with more public datasets, which will be provided in the final version of the paper.
>
> Table R2. Accuracy, fidelity, and ASR of different attacks across diverse public datasets.
>
> |Dataset|Method|Acc (\%)|Fid (\%)|ASR (\%)|
> |---|---|---|---|---|
> |Caltech256 (Training set)|Knockoff Nets|50.05|50.75|42.63|
> |Caltech256 (Training set)|SEEKER (ours)|65.71|66.54|78.26|
> |STL10 (Unlabeled set)|Knockoff Nets|78.74|79.84|52.36|
> |STL10 (Unlabeled set)|SEEKER (ours)|88.57|89.89|98.97|
>
>
> > Q2: For the different architectures in Table 3 and A2, what is the victim model architecture?
>
> We thank the reviewer for the question. The victim model architecture is ResNet-34 in Table 3 and A2 (changed to A3 for the revised version).
> We have also provided this information in Section 4.1 of our revised manuscript and Section I of our revised appendix.

---

### Official Review · Reviewer_c2Ta · 2023-11-04

**Soundness:** 3 good
**Presentation:** 3 good
**Contribution:** 3 good
**Rating:** 6
**Confidence:** 3

**Summary:**

The paper presents a query-efficient model extraction framework including query-free self-supervised training and query-efficient query generator. The proposed SEEKER method shows superior experimental results on multiple benchmark datasets.

**Strengths:**

The paper is well written and easy to understand. The idea of applying self0supervised training for model extraction is interesting.

**Weaknesses:**

The method itself, although works great, seems a bit ad-hoc. It will be great to see some theoretical justifications behind the framework.

**Questions:**

See weakness.

---

> ### Author Response · Authors · 2023-11-20
> **Response to Reviewer c2Ta**
>
> > Although the method works great, it will be great to see some theoretical justifications behind the framework.
>
> We really appreciate the positive comments and valuable suggestions of the reviewer.
> We would like to provide the following theoretical justifications for our proposed semantic alignment in this response.
> We have also included this content in Section C of our revised Appendix.
>
> The key insight of our semantic alignment approach is that the adversary can optimize a substitute model $S$ on a public dataset $\mathcal{D}\_{pub}$ to learn features that are similar to the victim model $V$, which is trained on a secret dataset $\mathcal{D}\_{secret}$.
> Without loss of generality, we assume the substitute model $S$ is composed of an encoding function $f$ and a fully connected layer $W$.
> In such a case, our observation can be reformulated as follows: if $f$ has a low semantic consistency loss and $W$ is trained on  $\mathcal{D}\_{secret}$ to evaluate the classification performance of $f$, $S$ has a low average classification loss on $\mathcal{D}\_{secret}$.
> To prove this observation, we first formally define the notations.
>
> The encoding function of $S$ belongs to $\mathcal{F}$, a class of representation functions $f: \mathcal{X} \rightarrow \mathbb{R}^d$, such that $||f(\cdot)||\leq R$ for some $R > 0$.
> We denote the set of all classes in $\mathcal{D}\_{pub}$ as $\mathcal{C}$, and each $c\in\mathcal{C}$ follows a probability distribution $\mathcal{D}\_c$.
> The supervised classification loss of $S$ on $\mathcal{D}\_{secret}$ can be defined as
> $$\mathcal{L}\_{\text{sup}}(S):= \mathbb{E}\_{(x,c)\in\mathcal{D}\_{secret}} \left[ l\left(S(x)\_c - S(x)\_{c'\neq c} \right) \right],$$
> where $l$ is a standard hinge loss or logistic loss, and $S=Wf$.
> When evaluating $S$, the best $W$ can be found by fixing $f$ and finetuning $W$. Therefore, we only denote the supervised loss of $f$ on $\mathcal{D}\_{secret}$ as:
> $$\mathcal{L}\_{\text{sup}}(f) =\inf_W \mathcal{L}\_{\text{sup}}(Wf).$$
> Also, our offline semantic consistency loss can be formalized as
> $$
>         \mathcal{L}\_C = -\mathbb{E}\_{x\in\mathcal{D}\_{pub}}\left[{\rm log}
>         \frac{{\rm sim}(S({\rm aug_{M}^1}(x)),S({\rm aug_{M}^2}(x)))}
>         {\sum_{x'\in\mathcal{D}\_{pub}, x' \neq x} {\rm sim}(S(x),S(x'))} \right],
> $$
> The loss term can be simplified as
> $$
> \mathcal{L}\_C  = \frac{1}{M} \sum_{i=1}^{M} l\left(f(x_j)^{T}(f(x_j^+) - f(x_j'))\right),
> $$
> Here, $x_j$ and $x_j^+$ are semantically equivalent data constructed by augmentations.
> With the notations above, we formalize a proposition as follows.
>
> **Proposition 1.**  *For a substitute model $S$ composed of an encoding function $f$ and a fully connected classification layer $W$, $S$ has a low average linear classification loss on $\mathcal{D}\_{secret}$ if $f$ has a low offline semantic consistency loss on $\mathcal{D}\_{pub}$.*
>
>
> We use a theorem proposed in [1] (Theorem 4.1 of the original paper) to prove this proposition.
> Let $\mathcal{S}=\lbrace x_j, x_j^+, x_j' \rbrace_{j=1}^M$ be the triplets sampled from $\mathcal{D}\_{pub}$ to optimize semantic consistency loss,
> $f\_{|\mathcal{S}} = \left(f_t(x_j), f_t(x_j^+), f_t(x_j')\right)\_{j\in[M],t\in[d]}\in \mathbb{R}^{3dM}$ be the restriction for $\mathcal{S}$ for any $f\in \mathcal{F}$, and we have a complexity measure with the following Rademacher average
> $$
> \mathcal{R}(\mathcal{F}) = \mathbb{E}\_{\sigma\in\lbrace\pm 1\rbrace^{3dM}}\left[\sup_{f\in \mathcal{F}}<\sigma, f\_{|\mathcal{S}}>\right].
> $$
> Let $\tau=\mathbb{E}\_{c,c'\sim\rho^2}\lbrace c=c'\rbrace$, and we have the following theorem[1]:
>
> **Theorem 1.** *With probability at least $1-\tau$, for all $f\in F$*
> $$\mathcal{L}\_{\text{sup}} (\widehat{f}) \leqslant \frac{1}{(1-\tau)} \mathcal{L}\_C(f) - \frac{\tau}{(1-\tau)} + \frac{1}{(1-\tau)} Gen_{M}$$
> *where*
> $$\ Gen_{M} = O\left(R \frac{R_{s}(F)}{M} + R^{2} \sqrt{\frac{\log \frac{1}{d}}{M}}\right)$$
> Here, we have $Gen_{M} \rightarrow 0$ as $M\rightarrow \infty$, and when $\rho$ is uniform and the number of classes $|C|\rightarrow \infty$, we have that $\frac{1}{(1-\tau)}\rightarrow 0, -\frac{\tau}{(1-\tau)}\rightarrow 0$.
> Therefore, when the number of sampled training triplets is large and $f$ has a low offline semantic consistency loss on $\mathcal{D}\_{pub}$, then $S$ has a low average linear classification loss on $\mathcal{D}\_{secret}$.
>
> [1] Saunshi, Nikunj, et al. "A theoretical analysis of contrastive unsupervised representation learning." International Conference on Machine Learning. PMLR, 2019.

---

### Author Response · Authors · 2023-11-20
**We appreciate the hard work of the reviewers and the chair.**

We really appreciate the hard work of the reviewers and the area chair.
We are encouraged that they found our idea interesting (R1) and novel (R3), our method clearly written (R1, R2),
and our experimental results strong (R2, R3) and thorough (R2).
We also thank the reviewers for informing us of what should be improved for representing our paper, such as providing more justification (R1), highlighting our contributions (R2), and providing more intuition for the design of the framework (R3).
We clarify some concerns and questions raised by each reviewer, and make modifications to our manuscript according to the valuable suggestions of the reviewers.

---

### Meta-Review · Area_Chair_mZrv · 2023-12-20

**Metareview:**

All reviewers found this paper interesting and appreciate the efforts that the authors put in during the rebuttal process. Their concerns centered around: (1) novelty of this work; (2) the design of the aggregator and (3) the source of empirical improvements.

**Justification For Why Not Higher Score:**

The reviewers have remaining concerns about (1) novelty of this work; (2) the design of the aggregator and (3) the source of empirical improvements.

**Justification For Why Not Lower Score:**

N/A

---

### Decision · Program_Chairs · 2024-01-16

Reject